# Expected Worst Case Regret via Stochastic Sequential Covering

**Changlong Wu**[1]                                                    *wuchangl@hawaii.edu*

**Mohsen Heidari**[1,2]                                                *mheidar@iu.edu*

**Ananth Grama**[1]                                                    *ayg@cs.purdue.edu*

**Wojciech Szpankowski**[1]                                            *szpan@purdue.edu*

[1] *Center for Science of Information (CSoI), Purdue University*
[2] *Department of Computer Science, Indiana University, Bloomington*

**Reviewed on OpenReview:** *https://openreview.net/forum?id=H1SekypXKA*

## Abstract

We study the problem of sequential prediction and online minimax regret with stochastically generated features under a general loss function. In an online learning setting, Nature selects features and associates a true label with these features. A learner uses features to predict a label, which is compared to the true label, and a loss is incurred. The total loss over $T$ rounds, when compared to a loss incurred by a set of experts, is known as a regret. We introduce the notion of *expected worst case minimax regret* that generalizes and encompasses prior known minimax regrets. For such minimax regrets, we establish tight upper bounds via a novel concept of *stochastic global sequential covering*. We show that for a hypothesis class of VC-dimension VC and *i.i.d.* generated features over $T$ rounds, the cardinality of stochastic global sequential covering can be upper bounded with high probability (w.h.p.) by $e^{O(\mathsf{VC} \cdot \log^2 T)}$. We then improve this bound by introducing a new complexity measure called the *Star-Littlestone* dimension, and show that classes with Star-Littlestone dimension SL admit a stochastic global sequential covering of order $e^{O(\mathsf{SL} \cdot \log T)}$. We further establish upper bounds for real valued classes with finite fat-shattering numbers. Finally, by applying information-theoretic tools for fixed design minimax regrets, we provide lower bounds for expected worst case minimax regret. We demonstrate the effectiveness of our approach by establishing tight bounds on the expected worst case minimax regrets for logarithmic loss and general mixable losses.

## 1 Introduction

Online learning (Shalev-Shwartz & Ben-David, 2014) can be viewed as a game between Nature and predictor/learner. At each time step $t$, Nature selects feature $\mathbf{x}_t \in \mathcal{X}$ and presents it to the learner. The learner then makes a prediction $\hat{y}_t \in \hat{\mathcal{Y}}$ based on history $\mathbf{x}^t = \{\mathbf{x}_1, \cdots, \mathbf{x}_t\}$ and $y^{t-1} = \{y_1, \cdots, y_{t-1}\}$, where $\mathbf{x}^t$ and $y^{t-1}$ are the features up to time $t$, and the true labels up to time $t-1$, respectively. Nature then reveals the true label $y_t \in \mathcal{Y}$ and the learner incurs some loss $\ell(\hat{y}_t, y_t)$, where we assume $\ell(\cdot, y_t)$ is convex for all $y_t \in \mathcal{Y}$. The game continues up to time $T$ and the goal is to minimize the pointwise regret defined as:

$$R(\hat{y}^T, y^T, \mathcal{H} \mid \mathbf{x}^T) = \sum_{t=1}^{T} \ell(\hat{y}_t, y_t) - \inf_{h \in \mathcal{H}} \sum_{t=1}^{T} \ell(h(\mathbf{x}_t), y_t),$$

where $\ell : \hat{\mathcal{Y}} \times \mathcal{Y} \to \mathbb{R}^+$ is a loss function and $\mathcal{H} \subset \hat{\mathcal{Y}}^{\mathcal{X}}$ is a class of *experts*. We say sequences $\mathbf{x}^T$ and $y^T$ are *realizable* if for some $h \in \mathcal{H}$ we have $h(\mathbf{x}_t) = y_t$ for $t \in [T]$, and *agnostic* otherwise.

Regret analysis for a general class $\mathcal{H}$ of experts is often studied via a sequential cover $\mathcal{G}$ of $\mathcal{H}$, which is defined as a set of functions mapping $\mathcal{X}^* \to \hat{\mathcal{Y}}$ (where $\mathcal{X}^*$ is the set of all finite sequences over $\mathcal{X}$) such that for all $h \in \mathcal{H}$

and $\mathbf{x}^T \in \mathcal{X}^T$, there exists $g \in \mathcal{G}$ satisfying $h(\mathbf{x}_t) = g(\mathbf{x}^t)$ for all $t \in [T]$. In the seminal work of Ben-David et al. (2009), the authors established a striking connection between the regrets of agnostic and realizable cases through the concept of sequential covering. One of the core arguments of Ben-David et al. (2009) is the following observation (see Lemma 2 for more formal assertion):

> *If a binary valued class $\mathcal{H}$ admits a predictor with cumulative error upper bounded by $B$ for realizable samples of length $T$, then $\mathcal{H}$ has a sequential cover $\mathcal{G}$ of size $O(T^{B+1})$.*

This is established by considering all the possible error patterns of the predictor in the realizable case. Using standard expert algorithms (e.g., Exponential Weighted Average), one then relates *agnostic* regrets to the size of $\mathcal{G}$. For example, for absolute loss, the regret bound is of the form $O(\sqrt{T \log |\mathcal{G}|}) = O(\sqrt{BT \log T})$, while for general mixable losses one finds $O(\log |\mathcal{G}|) = O(B \log T)$. In Ben-David et al. (2009), the authors derived upper bounds for $B$ through Littlestone dimension. This was further generalized in Daniely et al. (2011) to multi-label cases, and in Rakhlin et al. (2010) to the real valued case. However, all of these results assumed that features are presented adversarially. This may be too pessimistic and restrictive for modeling real scenarios of prediction, see e.g. (Rakhlin et al., 2011).

This paper generalizes the adversarial online learning setting into a more relaxed (and broader) stochastic scenario by restricting the adversary to generate the features $\mathbf{x}^T$ from some *unknown* distribution $\mu$ over $\mathcal{X}^T$ (not necessarily *i.i.d.*) in a known class $\mathcal{P}$ of distributions (i.e., the so called *universal* or *distribution blind* scenario). We introduce a novel general *expected worst case regret* defined as:

$$\tilde{r}_T(\mathcal{H}, \mathcal{P}, \phi^T) = \sup_{\mu \in \mathcal{P}} \mathbb{E}_{\mathbf{x}^T \sim \mu} \left[ \sup_{y^T} R(\hat{y}^T, y^T, \mathcal{H} \mid \mathbf{x}^T) \right],$$

where $\phi_t : \mathcal{X}^t \times \mathcal{Y}^{t-1} \to \hat{\mathcal{Y}}$ is any (deterministic) prediction rule (i.e., algorithm), and $\hat{y}_t = \phi_t(\mathbf{x}^t, y^{t-1})$. We also define the following expected worse case *minimax* regret:

$$\tilde{r}_T(\mathcal{H}, \mathcal{P}) = \inf_{\phi^T} \tilde{r}_T(\mathcal{H}, \mathcal{P}, \phi^T),$$

where $\phi^T$ runs over all possible (deterministic) prediction rules. The expected worst-case regret not only recovers previously known regrets by considering different classes of $\mathcal{P}$ (see Proposition 1), but also provides a new *analysis framework* for regret analysis in broader stochastic scenarios.

To capture the stochastic nature of the expected worst-case regret, we introduce a novel concept that we term *stochastic (global) sequential covering* that generalizes the classical (adversary) sequential covering. We say a class $\mathcal{G}$ of functions $\mathcal{X}^* \to \hat{\mathcal{Y}}$ is a *stochastic (global) sequential covering* of $\mathcal{H}$ w.r.t. $\mathcal{P}$ at scales $\alpha \geq 0$ and confidence $\delta > 0$, if for all $\mu \in \mathcal{P}$, we have:

$$\Pr_{\mathbf{x}^T \sim \mu} \left[ \exists h \in \mathcal{H} \, \forall g \in \mathcal{G} \exists t \in [T] \ s.t. \ |h(\mathbf{x}_t) - g(\mathbf{x}^t)| > \alpha \right] \leq \delta.$$

Similar to the adversarial case, we will show that the expected worst case minimax regret can be upper bounded by the size of a stochastic (global) sequential covering set $\mathcal{G}$ via standard expert algorithms. For instance, if $\mathcal{H}$ is binary valued and $\mathcal{G}$ is a stochastic sequential covering of $\mathcal{H}$ at scale $\alpha = 0$ and $\delta = \frac{1}{T}$, for absolute loss, we show that $\tilde{r}_T(\mathcal{H}, \mathcal{P}) = O(\sqrt{T \log |\mathcal{G}|})$, while for logarithmic loss and general mixable loss (e.g., square loss), we find $\tilde{r}_T(\mathcal{H}, \mathcal{P}) = O(\log |\mathcal{G}|)$; see Theorem 1 and 2 for more general statements of this fact.

Our goal is to derive tight regret bounds that go beyond the conventional $\sqrt{T}$ bounds for a general loss and provide explicit prediction rules (algorithms) that achieve such upper bounds. To retain focus, we will mainly consider the case when $\mathcal{P}$ is the class of all *i.i.d.* distributions over $\mathcal{X}^T$. We emphasize that our results also work for general *exchangeable* distributions (Aldous, 1985).

**Summary of main contributions.** We formulate the notion of *expected worst case regret* that generalizes and encompasses prior known online regret into a unified framework, which allows us to study broader classes of feature generating processes, e.g., the *distribution blind* (i.e., universal) scenarios. We provide a comprehensive analysis of the expected worst case regret when the processes are *i.i.d.* (and more generally the *exchangeable* processes) for a wide range of hypothesis classes and losses, via a novel concept of *stochastic sequential covering*. This allows us to derive tighter regret bounds and design new algorithms that achieve these bounds in both previously studied and

novel settings. Specifically, for binary valued class $\mathcal{H}$ of finite VC-dimension, we show that the stochastic sequential covering number is upper bounded w.h.p. by $O(\mathsf{VC}(\mathcal{H})\log^2 T)$ under $i.i.d.$ processes; this is tight upto a $\log T$ factor. Using this result we establish an $O(\mathsf{VC}(\mathcal{H})\log^2 T)$ regret bound for expert classes generated by *embedding* $\mathcal{H}$ into real valued classes under logarithmic loss. This improves substantially the $O(\sqrt{\mathsf{VC}(\mathcal{H})T})$ regret bound in Bhatt & Kim (2021), and subsumes the *average* (and well-specified) regrets implied by Bilodeau et al. (2021). We then extend this result in several directions. We show that the stochastic sequential covering number for binary valued classes is upper bounded by $O(\mathsf{SL}(\mathcal{H})\log T)$, where $\mathsf{SL}(\mathcal{H})$ is the *Star-Littlestone* dimension of $\mathcal{H}$ – a new dimension measure introduced in this paper. This bound is optimal on the logarithmic factor, and the technique for establishing this result is new in literature, and of independent interest. For small real valued classes $\mathcal{H}$ of finite Pseudo-dimension $\mathsf{P}(\mathcal{H})$, we show that the stochastic sequential covering at scale $\alpha$ is upper bounded w.h.p. by $O(\mathsf{P}(\mathcal{H})\log^2(T/\alpha))$. For general real valued class $\mathcal{H}$ with $\alpha$-fat-shattering numbers (i.e., the scale sensitive VC-dimension) of order $d(\alpha)$, we show that the stochastic sequential covering at scale $\alpha$ is upper bounded w.h.p. by $\tilde{O}(d(\alpha/32))$. Using this result, we show that the expected worst case regret under logarithmic loss is upper bounded by $\tilde{O}(T^{l/(l+1)})$ if $d(\alpha) = O(\alpha^{-l})$; this is tight up to a poly-logarithmic factor. Finally, we introduce a general approach for lower bounding the expected worst case regret via the *fixed-design* regret introduced in Wu et al. (2022b) that provides matching (upto poly-logarithmic factors) lower bounds for most of the our upper bounds presented above.

**Related work.**    Regret analysis of online learning problems dates back to the work of Littlestone & Warmuth (1994) and Vovk (1990), where the authors developed a general framework for the *Exponential Weighted Average* algorithm for finite expert classes. We refer to (Cesa-Bianchi & Lugosi, 2006) for an excellent discussion of this topic and its extensions. In Ben-David et al. (2009), the authors extended the framework to infinite classes with binary labels via the concept of sequential covering and subsequently generalized to the multi-class case in Daniely et al. (2011). In a series of papers Rakhlin et al. (2010); Rakhlin & Sridharan (2015); Rakhlin et al. (2015); Rakhlin & Sridharan (2014), the authors established a comprehensive framework for regret analysis of real valued classes via the concept of *sequential Rademacher complexity*. One of the core techniques in this line of work is to express regret in terms of an iterated minimax formulation, which is then be controlled by an expected majorizing of martingales via the minimax theorem. The latter is then computed using a *sequential covering* [1] number and the standard technique of chaining and Dudley integral. However, all of these efforts consider adversarial cases that may be too restrictive for real word scenarios.

In Lazaric & Munos (2009), the authors introduced a scenario where the features are generated by an unknown $i.i.d.$ source but the labels are still presented adversarially. In particular, Lazaric & Munos (2009) showed that for finite VC-dimensional classes and for absolute loss, regret grows as $O(\sqrt{\mathsf{VC}(\mathcal{H})T\log T})$. One of the core techniques in this work is an epoch approach that reduces infinite class to a finite class case using successive covering. However, their upper bound is dominated by a $\sqrt{T}$ term of the approximation error of covering that is too loose for many loss functions, e.g., logarithmic loss. Indeed, the same epoch approach (and its analysis of the approximation error) was used in Bhatt & Kim (2021) for logarithmic loss, resulting an $O(\sqrt{T})$ regret bound. In Bilodeau et al. (2021), the authors showed that for logarithmic loss and finite VC-dimensional classes, the *risk* grows as $O((\mathsf{VC}(\mathcal{H})\log^2 T)/T)$. However, their proof applies only to the *average case* minimax regret (see Section 2) and in the *realizable* (i.e., *well-specified*) case. In Rakhlin et al. (2011), the authors considered a scenario where at each time step Nature selects adversarially some distribution to sample, from a restricted class of distributions that are determined by previously generated samples (*not* previously selected distributions). This is characterized by the concept of *distribution dependent* Rademacher complexity, using a similar minimax approach as discussed above. However, their result only holds for the *distribution non-blind case* (i.e., when the distribution is known in advance), see (Rakhlin et al., 2011, Section 7). Note that all regrets analyzed in this paper are for the *distribution blind* case. We note also a recent line of research on *smooth adversaries* in Rakhlin et al. (2011); Haghtalab et al. (2020; 2022); Block et al. (2022) that share some technical similarity (e.g., symmetries of samples) with our work.

## 2   Problem Formulation

Let $\mathcal{X}$ be a feature space, $\mathcal{Y}$ be the true label space, and $\hat{\mathcal{Y}}$ be the space of outputs of the learner. We assume throughout the paper that $\hat{\mathcal{Y}} = [0,1]$. We denote by $\mathcal{H} \subset \hat{\mathcal{Y}}^{\mathcal{X}}$ a class of functions $\mathcal{X} \to \hat{\mathcal{Y}}$, which is also referred to as a hypothesis or experts class. For any time horizon $T$, we consider a class $\mathcal{P}$ of distributions over $\mathcal{X}^T$. We consider the following game between Nature and predictor. At the beginning of the game, Nature selects a distribution $\mu \in \mathcal{P}$ and samples

---

[1]Note that the sequential covering as in Rakhlin et al. (2010) is slightly different from the one we adopt in our work, since their definition relies on some underlying trees.

an input sequence $\mathbf{x}^T \sim \mu$, where $\mathbf{x}^T \in \mathcal{X}^T$. At each time step $t \leq T$, Nature reveals the $t$-th sample $\mathbf{x}_t$ of $\mathbf{x}^T$ to the predictor. The predictor then makes a prediction $\hat{y}_t \in \mathcal{Y}$ using a strategy $\phi_t : \mathcal{X}^t \times \mathcal{Y}^{t-1} \to \hat{\mathcal{Y}}$ potentially using the history observed thus far, that is, $\hat{y}_t = \phi_t(\mathbf{x}^t, y^{t-1})$. After the prediction, Nature reveals the true label $y_t$ and the predictor incurs a loss $\ell(\hat{y}_t, y_t)$ for some predefined convex loss function $\ell : \hat{\mathcal{Y}} \times \mathcal{Y} \to [0, \infty)$. We are interested in the following *expected worst case* minimax regret:

$$\tilde{r}_T(\mathcal{H}, \mathcal{P}) = \inf_{\phi^T} \sup_{\mu \in \mathcal{P}} \mathbb{E}_{\mathbf{x}^T \sim \mu} \left[ \sup_{y^T} \left( \sum_{t=1}^T \ell(\hat{y}_t, y_t) - \inf_{h \in \mathcal{H}} \sum_{t=1}^T \ell(h(\mathbf{x}_t), y_t) \right) \right], \tag{1}$$

where *worst case* indicates the predictor needs to compete with the best expert in $\mathcal{H}$ for any $\mathbf{x}^T$.

We now introduce two particular convex losses, the absolute loss and logarithmic loss as defined:

1. Let $\mathcal{Y} \subset \hat{\mathcal{Y}} = [0, 1]$, the absolute loss is defined as $\ell(\hat{y}, y) = |\hat{y} - y|$. Note that when $\mathcal{Y} = \{0, 1\}$, the absolute loss can be interpreted as the *expected* miss-classification loss, i.e., $\ell(\hat{y}, y) = \mathbb{E}_{b \sim \text{Bernoulli}(\hat{y})}[\mathbb{1}\{b \neq y\}]$, see (Shalev-Shwartz & Ben-David, 2014, Chapter 21);

2. Let $\mathcal{Y} = \{0, 1\}$ and $\hat{\mathcal{Y}} = [0, 1]$; the logarithmic loss (Log-loss) is defined as $\ell(\hat{y}, y) = -y \log(\hat{y}) - (1 - y) \log(1 - \hat{y})$. The Log-loss appears naturally in many different scenarios, e.g., sequential probability assignment and portfolio optimization, see (Cesa-Bianchi & Lugosi, 2006, Chapter 9 & 10).

We note that the expected worst case minimax regret $\tilde{r}_T(\mathcal{H}, \mathcal{P})$ recovers previously known minimax regrets by selecting appropriate distribution class $\mathcal{P}$. Indeed, in Shamir & Szpankowski (2021); Jacquet et al. (2021); Wu et al. (2022a), the following regrets are defined. The *fixed design* minimax regret for any given $\mathbf{x}^T \in \mathcal{X}^T$ is:

$$r_T^*(\mathcal{H} \mid \mathbf{x}^T) = \inf_{\phi^T} \sup_{y^T} \left( \sum_{t=1}^T \ell(\hat{y}_t, y_t) - \inf_{h \in \mathcal{H}} \sum_{t=1}^T \ell(h(\mathbf{x}_t), y_t) \right). \tag{2}$$

The *maximum* fixed design minimax regret is then: $r_T^*(\mathcal{H}) = \sup_{\mathbf{x}^T} r_T^*(\mathcal{H} \mid \mathbf{x}^T)$. Furthermore, the *sequential minimax regret* is

$$r_T^a(\mathcal{H}) = \inf_{\phi^T} \sup_{\mathbf{x}^T, y^T} \left( \sum_{t=1}^T \ell(\hat{y}_t, y_t) - \inf_{h \in \mathcal{H}} \sum_{t=1}^T \ell(h(\mathbf{x}_t), y_t) \right) \tag{3}$$

which is equivalent[2] to the iterated minimax regret as in Rakhlin et al. (2010).

We also introduce the following expected *average case* minimax regret:

$$\bar{r}_T(\mathcal{H}, \mathcal{P}) = \inf_{\phi^T} \sup_{\mu \in \mathcal{P}, h \in \mathcal{H}} \mathbb{E}_{\mathbf{x}^T \sim \mu} \left[ \sup_{y^T} \left( \sum_{t=1}^T \ell(\hat{y}_t, y_t) - \ell(h(\mathbf{x}_t), y_t) \right) \right] \tag{4}$$

where the main difference with $\tilde{r}_T(\mathcal{H}, \mathcal{P})$ is the position of $\sup_h$. Note that this concept subsumes the setups of Bhatt & Kim (2021); Bilodeau et al. (2021) except that the authors of Bhatt & Kim (2021); Bilodeau et al. (2021) consider a weaker *well-specified* setting for generating $y^T$.

The following observation is easy to prove and shows that $\tilde{r}_T$ is indeed a more general concept:

**Proposition 1.** *If $\mathcal{P}$ is a class of all singleton distributions over $\mathcal{X}^T$, then $\tilde{r}_T(\mathcal{H}, \mathcal{P}) = r_T^a(\mathcal{H})$ for all $\mathcal{H}$. If $\mathcal{P}$ is the singleton distribution that assigns probability 1 for $\mathbf{x}^T$, then $\tilde{r}_T(\mathcal{H}, \mathcal{P}) = r_T^*(\mathcal{H} \mid \mathbf{x}^T)$. Furthermore, $\tilde{r}_T(\mathcal{H}, \mathcal{P}) \geq \bar{r}_T(\mathcal{H}, \mathcal{P})$, for any $\mathcal{H}$ and $\mathcal{P}$.*

**Example 1.** *To understand differences between $\tilde{r}_T$ and $\bar{r}_T$, we consider the following example. Let $\mathcal{H}$ be the class of all functions $[0, 1] \to \{0, 1\}$ that takes value 1 on at most $T$ positions and 0 otherwise. Let $\nu$ be the uniform distribution over $[0, 1]$, and $\ell(\hat{y}_t, y_t) = |\hat{y}_t - y_t|$, where $\hat{y}_t \in [0, 1]$ and $y_t \in \{0, 1\}$. We will denote by $\nu^T$ the i.i.d distribution of length $T$ with marginal $\nu$. We have $\bar{r}_T(\mathcal{H}, \{\nu^T\}) = 0$, since for any $h$, w.p. 1 we have $h(x_t) = 0$ for all $t \in [T]$, meaning that a strategy that predicts 0 all the time incurs 0 regret. However, we also have $\tilde{r}_T(\mathcal{H}, \{\nu^T\}) \geq \frac{T}{2}$. To see this, we choose $y^T \in \{0, 1\}^T$ uniformly at random and observe that any strategy will make at least $\frac{T}{2}$ accumulated losses, however, for any $\mathbf{x}^T$ and $y^T$, there exists $h \in \mathcal{H}$ such that $\forall t \in [T]$, $h(\boldsymbol{x}_t) = y_t$.*

---

[2]The equivalence is a well-known result, see e.g., (Cesa-Bianchi & Lugosi, 2006, Exercise 2.18) or (Wu et al., 2022a, Lemma 2).

**Remark 1.** *We should remark that our definition of both $\tilde{r}_T$ and $\bar{r}_T$ are distribution blind in the sense of (Rakhlin et al., 2011, Section 7), since the dependency of marginals of $\mu$ is not just through previously generated samples as in Rakhlin et al. (2011). For instance, suppose $\mathcal{P}$ is the class of all i.i.d. processes, one may express the regret in the following iterated minimax manner:*

$$\sup_{\nu} \mathbb{E}_{\boldsymbol{x}_1 \sim \nu} \inf_{\hat{y}_1} \sup_{y_1} \cdots \mathbb{E}_{\boldsymbol{x}_T \sim \nu} \inf_{\hat{y}_T} \sup_{y_T} R(\hat{y}^T, y^T, \mathcal{H} \mid \boldsymbol{x}^T).$$

*We note that this expression implicitly assumed that the distribution $\nu$ is known to the learner (otherwise the optimization function for each $\inf_{\hat{y}_t}$ is not well defined).*

## 3 Stochastic Sequential Cover

Let $\mathcal{X}^*$ be the set of all finite sequences over $\mathcal{X}$. We introduce one of our main technical ingredient of this paper, i.e., the stochastic (global) sequential covering, as follows:

**Definition 1** (Stochastic sequential cover). *We say a class $\mathcal{G}$ of functions $\mathcal{X}^* \to [0,1]$ is a stochastic global sequential cover of a class $\mathcal{H} \subset [0,1]^{\mathcal{X}}$ w.r.t. the class $\mathcal{P}$ of distributions over $\mathcal{X}^T$ at scale $\alpha > 0$ and confidence $\delta > 0$, if for all $\mu \in \mathcal{P}$, we have*

$$\Pr_{\boldsymbol{x}^T \sim \mu} \left[ \exists h \in \mathcal{H} \, \forall g \in \mathcal{G} \, \exists t \in [T] \text{ s.t. } |h(\boldsymbol{x}_t) - g(\boldsymbol{x}^t)| > \alpha \right] \leq \delta.$$

*We define the minimal size of $\mathcal{G}$ to be the stochastic global sequential covering number of $\mathcal{H}$.*

Note that the distribution class $\mathcal{P}$ in Definition 1 is completely general and recovers the classical sequential covering as in Ben-David et al. (2009); Rakhlin et al. (2010) if $\mathcal{P}$ is the class of all singleton distributions over $\mathcal{X}^T$. Likewise, all the results established in this section hold for *any* distribution class $\mathcal{P}$.

To begin with, we first establish the following simple (but useful) composition property of stochastic sequential cover. Let $\mathcal{H}_1, \cdots, \mathcal{H}_m \subset [0,1]^{\mathcal{X}}$ be $m$ function classes over the same domain and $\Theta$ be a parameter space equipped with some norm $||\cdot||$. For any function $F : [0,1]^m \times \Theta \to [0,1]$ such that $\forall \mathbf{z}_1, \mathbf{z}_2 \in [0,1]^m$, $\boldsymbol{\theta}_1, \boldsymbol{\theta}_2 \in \Theta$ we have $F(\mathbf{z}_1, \boldsymbol{\theta}_1) - F(z_2, \boldsymbol{\theta}_2) \leq L \max\{||\mathbf{z}_1 - \mathbf{z}_2||_{\infty}, ||\boldsymbol{\theta}_1 - \boldsymbol{\theta}_2||\}$ for some constant $L \in \mathbb{R}^+$, the $F$-composition of $\mathcal{H}_1, \cdots, \mathcal{H}_m$ and $\Theta$ is defined to be the class:

$$\mathcal{H} = \{h(\mathbf{x}) = F(h_1(\mathbf{x}), \cdots, h_m(\mathbf{x}), \boldsymbol{\theta}) : \forall i \in [m], \ h_i \in \mathcal{H}_i \text{ and } \boldsymbol{\theta} \in \Theta\}.$$

**Proposition 2.** *Let $\mathcal{H}_1, \cdots, \mathcal{H}_m \subset [0,1]^{\mathcal{X}}$ be any classes, $\Theta$ be any parameter space equipped with norm $||\cdot||$, and $F$ be any function satisfying the conditions above. If $\forall i \in [m]$, $\mathcal{H}_i$ admits a statistical sequential covering set $\mathcal{G}_i$ at scale $\alpha/L$ and confidence $\delta/m$ w.r.t. distribution class $\mathcal{P}$, and $\Theta$ admits an $\alpha/L$ cover $\mathcal{C}$ under norm $||\cdot||$, then the $F$-composition class $\mathcal{H}$ of $\mathcal{H}_1, \cdots, \mathcal{H}_m$ and $\Theta$ admits a statistical sequential covering set $\mathcal{G}$ w.r.t. $\mathcal{P}$ at scale $\alpha$ and confidence $\delta$ such that:*

$$|\mathcal{G}| \leq |\mathcal{C}| \prod_{i=1}^{m} |\mathcal{G}_i|.$$

*Proof.* For any tuple of indices $(j_1, \cdots, j_m)$ with $j_i \in [|\mathcal{G}_i|]$ and $\boldsymbol{\theta}' \in \mathcal{C}$, we construct a function $g$ such that:

$$g(\mathbf{x}^t) = F(g_{j_1}(\mathbf{x}^t), \cdots, g_{j_m}(\mathbf{x}^t), \boldsymbol{\theta}'),$$

where $g_{j_i}$ is the $j_i$th function in $\mathcal{G}_i$. The covering set $\mathcal{G}$ is defined to be the class containing of all such functions $g$. For any function $h \in \mathcal{H}$, there exist $h_1, \cdots, h_m$ with $h_i \in \mathcal{H}_i$ and $\boldsymbol{\theta} \in \Theta$ such that for all $\mathbf{x} \in \mathcal{X}$, $h(\mathbf{x}) = F(h_1(\mathbf{x}), \cdots h_m(\mathbf{x}), \boldsymbol{\theta})$. By union bound and definition of stochastic sequential covering of $\mathcal{G}_i$, w.p. $\geq \delta$ over $\mathbf{x}^T$, for all $i \in [m]$, there exist $g_{j_i} \in \mathcal{G}_i$ such that $\forall t \in [T]$, $|g_{j_i}(\mathbf{x}^t) - h_i(\mathbf{x}_t)| \leq \alpha/L$. One can verify that the function $g$ corresponding to $(j_1, \cdots, j_m)$ and $\boldsymbol{\theta}' \in \mathcal{C}$ closest to $\boldsymbol{\theta}$ under $||\cdot||$ is the desired function that covers $h$ on $\mathbf{x}^T$, due to the $L$-Lipschitz property of $F$. $\square$

We provide several examples below that demonstrate how $F$-composition can be exploited to generate interesting complex classes from simple classes.

**Example 2.** *Let $\Theta = [0,1]^2$ and $\mathcal{H}_1 \subset \{0,1\}^{\mathcal{X}}$ be a binary valued class of finite VC-dimension. If we take $F(y,\boldsymbol{\theta}) = y\theta_1 + (1-y)\theta_2$ for $y \in \{0,1\}$ and $\boldsymbol{\theta} \in [0,1]^2$, the F-composition class $\mathcal{H} \subset [0,1]^{\mathcal{X}}$ of $\mathcal{H}_1$ and $\Theta$ recovers the setup of Bhatt & Kim (2021). We note that in this case the set $\Theta$ admits an $\alpha$-covering set of size $O(\alpha^{-2})$ under $L_\infty$ norm for all $\alpha > 0$ and $F$ is 1-Lipschitz in the sense of Proposition 2.*

**Example 3.** *Let $\Theta = \{\theta_1 + \cdots + \theta_d \leq 1 : \boldsymbol{\theta} \in [0,1]^d\}$ for some $d \in \mathbb{N}^+$ and $\mathcal{H}_1, \cdots, \mathcal{H}_d \subset \{0,1\}^{\mathcal{X}}$ be $d$ binary valued classes of finite VC-dimension. If we take $F(\mathbf{y}^d, \boldsymbol{\theta}) = \langle \boldsymbol{\theta}, \mathbf{y}^d \rangle$ for $\mathbf{y}^d \in \{0,1\}^d$ and $\boldsymbol{\theta} \in \Theta$, the F-composition class $\mathcal{H} \subset [0,1]^{\mathcal{X}}$ of $\mathcal{H}_i$s and $\Theta$ defines a natural class. We note that in this case $\Theta$ is $\alpha$-covered by a set of size $\alpha^{-d}$ under $L_1$ norm and $F$ is 1-Lipschitz in the sense of Proposition 2. Moreover, if we take $d = 2$ and $\mathcal{H}_2 = \{1 - h(\boldsymbol{x}) : h \in \mathcal{H}_1\}$ we subsume the setup of Example 2.*

**Example 4.** *Let $\Theta$ be empty, $\mathcal{X} = \mathbb{R}^d$ and $\mathcal{H}_i = \{h_{[a,b]}(\boldsymbol{x}) = 1\{\boldsymbol{x}[i] \in [a,b]\} : [a,b] \subset [0,1], \boldsymbol{x} \in \mathbb{R}^d\}$ for $i \in [d]$, i.e., $\mathcal{H}_i$ is the class of indicators of interval on the ith coordinate of $\boldsymbol{x}$. If we take $F(y_1, \cdots, y_d) = \prod_{i=1}^d y_i$ for $\mathbf{y}^d \in \{0,1\}^d$, the F-composition class $\mathcal{H} \subset \{0,1\}^{\mathcal{X}}$ of $\mathcal{H}_i$s defines the class of indicators of rectangular cuboids in $\mathbb{R}^d$ and $F$ is 1-Lipschitz.*

### 3.1 Upper bounds on regret via stochastic sequential covering

We now prove two general results below that demonstrate how a bound on the stochastic sequential covering number implies bounds on the expected worst case regret $\tilde{r}_T$ in an algorithmic fashion.

**Theorem 1.** *Let $\mathcal{H}$ be a set of functions mapping $\mathcal{X} \to [0,1]$ and $\mathcal{G}_\alpha$ be a stochastic global sequential covering of $\mathcal{H}$ at scale $\alpha$ and confidence $\delta = 1/T$ w.r.t. distribution class $\mathcal{P}$. If $\ell(\cdot, y)$ is convex, L-Lipschitz and bounded by 1 on $\hat{\mathcal{Y}}$ for any $y \in \mathcal{Y}$, then:*

$$\tilde{r}_T(\mathcal{H}, \mathcal{P}) \leq \inf_{0 \leq \alpha \leq 1} \left\{ \alpha L T + \sqrt{(T/2) \log |\mathcal{G}_\alpha|} + 1 \right\}.$$

*If, in addition, $\ell$ is $\eta$-Mixable (Cesa-Bianchi & Lugosi, 2006, Chapter 3.5) then:*

$$\tilde{r}_T(\mathcal{H}, \mathcal{P}) \leq \inf_{0 \leq \alpha \leq 1} \left\{ \alpha L T + \frac{1}{\eta} \log |\mathcal{G}_\alpha| + 1 \right\}.$$

*Proof.* We run the Exponential Weighted Average (EWA) algorithm (Cesa-Bianchi & Lugosi, 2006, Page 14) on $\mathcal{G}_\alpha$. We split the regret into two parts, one that is incurred by the predictor against $\mathcal{G}_\alpha$ and the other that is incurred by the discrepancy between $\mathcal{G}_\alpha$ and $\mathcal{H}$. For the first term, we have by standard result (Cesa-Bianchi & Lugosi, 2006, Theorem 2.2) that with probability 1 on $\mathbf{x}^T$:

$$\sum_{t=1}^{T} \ell(\hat{y}_t, y_t) \leq \inf_{g \in \mathcal{G}_\alpha} \sum_{t=1}^{T} \ell(g(\mathbf{x}^t), y_t) + \sqrt{(T/2) \log |\mathcal{G}_\alpha|}.$$

For the second term, we denote by $A$ the event described in the probability of Definition 1. Since $\Pr[A] \leq \frac{1}{T}$ and $\ell(\hat{y}, y) \leq 1$ by assumption, the expected regret contributed by the event $A$ is at most 1. We now condition on the event that $A$ does not happen. By Definition 1, we obtain $\forall h \in \mathcal{H} \exists g \in \mathcal{G}_\alpha \forall t \in [T], |h(\mathbf{x}_t) - g(\mathbf{x}^t)| \leq \alpha$. Since $\ell$ is bounded by 1 and $L$-Lipschitz, we have:

$$\inf_{h \in \mathcal{H}} \sum_{t=1}^{T} \ell(h(\mathbf{x}_t), y_t) \geq \inf_{g \in \mathcal{G}_\alpha} \sum_{t=1}^{T} \ell(g(\mathbf{x}^t), y_t) - \alpha L T.$$

The result follows by combining these inequalities.

For $\eta$-mixable loss, we replace the EWA algorithm with the Aggregation Algorithm of Cesa-Bianchi & Lugosi (2006, Chapter 3.5) and apply the result in Cesa-Bianchi & Lugosi (2006, Proposition 3.2). □

**Remark 2.** *We assume the loss to be convex for clarity of presentation. However, the result in Theorem 1 can be easily extended to bounded non-convex Lipschitz losses by replacing the EWA algorithm with its randomized counterpart as in Cesa-Bianchi & Lugosi (2006, Chapter 4.2).*

**Theorem 2.** *Let $\mathcal{Y} = \{0, 1\}$, $\hat{\mathcal{y}} = [0, 1]$ and $\ell$ be the logarithmic loss. If for all $\alpha \geq 0$ there exists a stochastic sequential covering set $\mathcal{G}_\alpha$ of class $\mathcal{H} \subset [0, 1]^{\mathcal{X}}$ w.r.t. distribution class $\mathcal{P}$ at scale $\alpha$ and confidence $\delta = \frac{1}{T}$, then:*

$$\tilde{r}_T(\mathcal{H}, \mathcal{P}) \leq \inf_{0 \leq \alpha \leq 1} \{2\alpha T + \log(|\mathcal{G}_\alpha| + 1) + \log(|\mathcal{G}_\alpha| + 1)/T + 1\}.$$

*Proof.* The proof is similar to the proof of Theorem 1, but replacing the EWA algorithm with the Smooth truncated Bayesian Algorithm introduced recently in Wu et al. (2022b) and running the algorithm on $\mathcal{G}_\alpha \cup \{u\}$ with truncation parameter $\alpha$ and uniform prior, where $u$ is the function that maps to $\frac{1}{2}$ for all $\mathbf{x}^t$. We again split the regret into two parts, one incurred by the Smooth truncated Bayesian Algorithm, and the other incurred by the error of covering. By Wu et al. (2022b, Theorem 1), the first term is upper bounded by $2\alpha T + \log(|\mathcal{G}_\alpha| + 1)$. For the error term, we note that we have added all $\frac{1}{2}$ valued functions $u$ into the expert class when running the Smooth truncated Bayesian Algorithm. This implies that the prediction rule can only incur the *actual* accumulated losses upper bounded by $T + \log(|\mathcal{G}_\alpha| + 1)$. Therefore, when the event $A$ (defined in Theorem 1) happens, the expected regret only contributes $(T + \log(|\mathcal{G}_\alpha| + 1)) \cdot \Pr[A] \leq (T + \log(|\mathcal{G}_\alpha| + 1))/T$. The result follows by combining the inequalities. $\square$

## 4 Stochastic Cover for Binary Valued Classes

This section focuses on the stochastic sequential covering number of binary valued classes. We assume that $\mathcal{P}$ is the class of all *i.i.d.* distributions over $\mathcal{X}^T$; however, our results hold for general *exchangeable* processes (Aldous, 1985) over $\mathcal{X}^T$ as well, i.e., distributions that are invariant under permutation of the indexes.

Note that the selection of $\mathcal{H}$ being binary valued is for clarity of exposition for bounding the stochastic covering number, since binary valued classes present arguably the simplest case yet provide enough structure to develop interesting insights. It is easy to extend the results established in this section to the real valued case via the composition property as in Proposition 2.

### 4.1 Stochastic sequential cover for finite VC-class

Let $\mathcal{H} \subset \{0, 1\}^{\mathcal{X}}$ be binary valued class with finite VC-dimension. We write $\mathsf{VC}(\mathcal{H})$ for the VC-dimension of $\mathcal{H}$. We show that the stochastic global sequential covering number can be upper bounded by $e^{O(\mathsf{VC}(\mathcal{H})\log^2 T)}$ w.h.p. using the *1-inclusion graph* algorithm that was introduced in Haussler et al. (1994). Without going into the technical details of the *1-inclusion graph* algorithm, we can understand it as a function that maps $(\mathcal{X} \times \{0, 1\})^{t-1} \times \mathcal{X} \to \{0, 1\}$, for any given $t \geq 1$. For $\mathcal{H}$ of finite VC-dimension and any function $\Phi : (\mathcal{X} \times \{0, 1\})^{t-1} \times \mathcal{X} \to \{0, 1\}$, we define the following quantity (here, we follow the notation in Haussler et al. (1994)):

$$\hat{M}_{\Phi, \mathcal{H}}(t) = \sup_{\mathbf{x}^t \in \mathcal{X}^t} \sup_{h \in \mathcal{H}} \mathbb{E}_\sigma \left[ 1\{\Phi(\mathbf{x}^{\sigma(t)}, h(\{\mathbf{x}^{\sigma(t-1)}\})) \neq h(\mathbf{x}_{\sigma(t)})\} \right],$$

where $\sigma$ is the uniform random permutation over $[t]$, $\mathbf{x}^{\sigma(t)} = \{\mathbf{x}_{\sigma(1)}, \cdots, \mathbf{x}_{\sigma(t)}\}$ and $h(\{\mathbf{x}^{\sigma(t-1)}\}) = \{h(\mathbf{x}_{\sigma(1)}), \cdots, h(\mathbf{x}_{\sigma(t-1)})\}$. The main result of Haussler et al. (1994) is stated as follows:

**Theorem 3** (Haussler et al., Theorem 2.3(ii))**.** *For any binary valued class $\mathcal{H}$ of finite VC-dimension and for any $t \geq 1$, there exists a function $\Phi : (\mathcal{X} \times \{0, 1\})^{t-1} \times \mathcal{X} \to \{0, 1\}$, i.e., the 1-inclusion graph algorithm, that satisfies*

$$\hat{M}_{\Phi, \mathcal{H}}(t) \leq \frac{\mathsf{VC}(\mathcal{H})}{t}.$$

Our main result for this part is as follows, with the proof presented below Lemma 2.

**Theorem 4.** *For any binary valued class $\mathcal{H}$ with finite VC-dimension, there exists a global sequential covering set $\mathcal{G}$ of $\mathcal{H}$ w.r.t. the class of all $i.i.d.$ distributions over $\mathcal{X}^T$ at scale $\alpha = 0$ and confidence $\delta$ such that for $T \geq e^5$ we have:*

$$\log|\mathcal{G}| \leq 5\mathsf{VC}(\mathcal{H})\log^2 T + \log T \log(1/\delta) + \log T.$$

The main idea for proving Theorem 4 is to show that for the 1-inclusion graph predictor $\Phi$, we have w.p. $\geq 1 - \delta$ over the sample $\mathbf{x}^T \overset{i.i.d}{\sim} \mu$, the *cumulative* error is upper bounded by $O(\mathsf{VC}(\mathcal{H})\log T + \log(1/\delta))$. Assuming this holds, one will be able to construct the covering functions in a similar fashion as Ben-David et al. (2009, Lemma 12).

The bound will follow by counting the error patterns. However, a direct application of Theorem 3 will only give us an *expected* $\mathsf{VC}(\mathcal{H})\log T$ error bound. The main challenge follows from the fact that even though the samples $\mathbf{x}^T$ are generated *i.i.d.*, the predictions made by the 1-inclusion predictor are *not* independent (neither a martingale), and therefore the standard concentration inequalities do not apply directly.

Our main proof technique exploits *permutation invariance* of the 1-inclusion graph predictor, which allows us to relate the cumulative error to a *reversed* martingale[3]. Using Bernstein inequality for martingales, we then establish the following key lemma, see Appendix A for a detailed proof.

**Lemma 1.** *Let* $\Phi : (\mathcal{X} \times \{0,1\})^* \times \mathcal{X} \to \{0,1\}$ *and* $h : \mathcal{X} \to \{0,1\}$ *be functions such that* $\Phi$ *is permutation invariant on* $(\mathcal{X} \times \{0,1\})^*$. *If for all* $t \in [T]$ *and* $\mathbf{x}^t \in \mathcal{X}^t$ *we have:*

$$\Pr\nolimits_{\sigma_t} \left[ \Phi(\mathbf{x}^{\sigma_t(t)}, h(\{\mathbf{x}^{\sigma_t(t-1)}\})) \neq h(\mathbf{x}_{\sigma_t(t)}) \right] \leq \frac{C}{t},$$

*where* $\sigma_t$ *is the uniform random permutation on* $[t]$ *and* $C \in \mathbb{N}^+$, *then for all* $\delta > 0$ *and* $T \geq e^5$

$$\Pr\nolimits_{\sigma_T} \left[ \sum_{t=1}^{T} \mathbb{1}\{\Phi(\mathbf{x}^{\sigma_T(t)}, h(\{\mathbf{x}^{\sigma_T(t-1)}\})) \neq h(\mathbf{x}_{\sigma_T(t)})\} \geq 4C\log T + \log(1/\delta) \right] \leq \delta.$$

**Lemma 2** (From error bound to covering). *Let* $\mathcal{H} \subset \{0,1\}^{\mathcal{X}}$ *be a binary valued class and* **err** $\in \mathbb{N}^+$. *For any* $\Omega \subset \mathcal{X}^T$, *suppose there exists a prediction rule* $\Phi$ *such that* $\forall h \in \mathcal{H}$, $\forall \mathbf{x}^T \in \Omega$, $\sum_{t=1}^{T} \mathbb{1}\{\Phi(\mathbf{x}^t, h(\{\mathbf{x}^{t-1}\})) \neq h(\mathbf{x}_t)\} \leq$ **err**. *Then, there exists a covering set* $\mathcal{G} \subset \{0,1\}^{\mathcal{X}^*}$ *such that for all* $\mathbf{x}^T \in \Omega$ *and* $h \in \mathcal{H}$ *one can find* $g \in \mathcal{G}$ *that satisfies* $g(\mathbf{x}^t) = h(\mathbf{x}_t)$ *for all* $t \in [T]$, *and*

$$|\mathcal{G}| \leq \sum_{t=0}^{err} \binom{T}{t} \leq T^{err+1}.$$

*Proof.* For any $I \subset [T]$ with $|I| \leq$ **err**, we construct a function $g_I$ as follows. Let $\mathbf{x}^t$ be the input, if $t \in I$, we set $g_I(\mathbf{x}^t) = 1 - \Phi(\mathbf{x}^t, g_I(\{\mathbf{x}^i\}_{i=1}^{t-1}))$, else, we set $g_I(\mathbf{x}^t) = \Phi(\mathbf{x}^t, g_I(\{\mathbf{x}^i\}_{i=1}^{t-1}))$ where $g_I(\{\mathbf{x}^i\}_{i=1}^{t-1}) = \{g_I(\mathbf{x}^1), \cdots, g_I(\mathbf{x}^{t-1})\}$. We claim that the set $\mathcal{G}$ that consists of all such $g_I$s is the desired covering set. To see this, for any $h \in \mathcal{H}$ and $\mathbf{x}^T \in \Omega$ we have $\sum_{t=1}^{T} \mathbb{1}\{\Phi(\mathbf{x}^t, h(\{\mathbf{x}^{t-1}\})) \neq h(\mathbf{x}_t)\} \leq$ **err**. Let $I$ be the positions $i \in [T]$ for which $\Phi(\mathbf{x}^i, h(\{\mathbf{x}^{i-1}\})) \neq h(\mathbf{x}_i)$, where $|I| \leq$ **err**. By construction, it is easy to check that $\forall t \in [T]$, $g_I(\mathbf{x}^t) = h(\mathbf{x}_t)$. The upper bound for $|\mathcal{G}|$ follows by counting the number of $I$s. See (Ben-David et al., 2009, Lemma 12). $\qquad\square$

*Proof of Theorem 4.* Let $\Phi$ be the 1-inclusion graph predictor. We have that $\Phi$ is permutation invariant, since the nodes in the 1-inclusion graph are determined by subsets of $\mathcal{X}$ that do not depend on the order of elements in the set. By symmetries of *i.i.d.* distributions, for any event $A(\mathbf{x}^T)$ on $\mathbf{x}^T \overset{i.i.d.}{\sim} \mu$, we have:

$$\Pr[A(\mathbf{x}^T)] = \mathbb{E}_\sigma[\Pr\nolimits_{\mathbf{x}^T}[A(\mathbf{x}_{\sigma(1)}, \cdots, \mathbf{x}_{\sigma(T)})]] = \mathbb{E}_\sigma \mathbb{E}_{\mathbf{x}^T} \mathbb{1}\{A(\mathbf{x}_{\sigma(1)}, \cdots, \mathbf{x}_{\sigma(T)})\} = \mathbb{E}_{\mathbf{x}^T} \mathbb{E}_\sigma \mathbb{1}\{(A(\mathbf{x}_{\sigma(1)}, \cdots, \mathbf{x}_{\sigma(T)})\}$$
$$\leq \sup_{\mathbf{x}^T} \Pr\nolimits_\sigma[A(\mathbf{x}_{\sigma(1)}, \cdots, \mathbf{x}_{\sigma(T)})],$$

where the interchange of the expectations follows from Fubini's theorem. It is therefore sufficient to show that for any $\mathbf{x}^T \in \mathcal{X}^T$, w.p. $\geq 1 - \delta$ over a random permutation $\sigma$ on $[T]$,

$$\sup_{h \in \mathcal{H}} \sum_{t=1}^{T} \mathbb{1}\{\Phi(\mathbf{x}^{\sigma(t)}, h(\{\mathbf{x}^{\sigma(t-1)}\})) \neq h(\mathbf{x}_{\sigma(t)})\} \leq 5\mathsf{VC}(\mathcal{H})\log T + \log(1/\delta).$$

To see this, we observe that by Sauer's lemma (Shalev-Shwartz & Ben-David, 2014), there are at most $T^{\mathsf{VC}(\mathcal{H})}$ functions of $\mathcal{H}$ restricted on any given $\mathbf{x}^T$. Let now $\delta$ in Lemma 1 be $\frac{\delta}{T^{\mathsf{VC}(\mathcal{H})}}$ and $C = \mathsf{VC}(\mathcal{H})$. When applying Theorem 3 together with a union bound, the error bound w.p. $\geq 1 - \delta$ is of the form $5\mathsf{VC}(\mathcal{H})\log T + \log(1/\delta)$. The upper bound for the size of covering set $\mathcal{G}$ follows from Lemma 2 by taking $\Omega \subset \mathcal{X}^T$ to be the set for which $\Phi$ makes at most $5\mathsf{VC}(\mathcal{H})\log T + \log(1/\delta)$ accumulated errors, where $\Pr[\Omega] \geq 1 - \delta$. $\qquad\square$

---

[3]Note that Vovk et al. (2005, Proposition 10.2) also considers a similar martingale based approach only for an almost sure rate.

Theorem 4 and Theorem 1 immediately imply the following regret bound.

**Corollary 1.** *Let $\mathcal{H} \subset \{0,1\}^{\mathcal{X}}$ be a binary valued class with finite VC-dimension, $\mathcal{P}$ be the class of all i.i.d. distributions over $\mathcal{X}^T$ and $T \geq e^5$. If $\ell(\cdot, y)$ is convex, L-lipschitz and bounded by $1$ for all $y \in \mathcal{Y}$, then:*

$$\tilde{r}_T(\mathcal{H}, \mathcal{P}) \leq \sqrt{3T\mathsf{VC}(\mathcal{H})\log^2 T} + O(1).$$

*If in addition $\ell$ is $\eta$-Mixable then*

$$\tilde{r}_T(\mathcal{H}, \mathcal{P}) \leq \frac{6}{\eta}\mathsf{VC}(\mathcal{H})\log^2 T + O(1).$$

The first bound of Corollary 1 recovers Lazaric & Munos (2009) but with a worse $\log T$ term. However, our result establishes the (essentially) same result by using a completely different technique. Moreover, our technique can be applied to more general problems than the epoch based approach of Lazaric & Munos (2009). Indeed, our $\log^2 T$ sequential covering bound is the key to finding a $O(\log^2 T)$ regret for mixable losses while the analysis of Lazaric & Munos (2009) can only give a $O(\sqrt{T})$ bound.

For logarithmic loss, we have the following regret bound:

**Corollary 2.** *Let $\mathcal{H}$ be a F-composition class of $\mathcal{H}_1, \cdots, \mathcal{H}_d \subset \{0,1\}^{\mathcal{X}}$ with $\Theta$ as in Example 3, $\mathcal{P}$ be the class of all i.i.d. distributions over $\mathcal{X}^T$, and $T \geq e^5$. If $\ell$ is the logarithmic loss, then:*

$$\tilde{r}_T(\mathcal{H}, \mathcal{P}) \leq O\left(\left(d + \sum_{i=1}^{d}\mathsf{VC}(\mathcal{H}_i)\right)\log^2 T\right).$$

*Proof.* Taking $\alpha = \frac{1}{T}$, we note that $\Theta$ can be $\alpha$-covered by a set $\mathcal{C}$ of size upper bounded by $T^d$ under $L_1$ norm. Applying Proposition 2 and Theorem 4 and noticing that the composition function $F$ is 1-Lipschitz, there exists a stochastic sequential covering set $\mathcal{G}$ of $\mathcal{H}$ w.r.t. i.i.d. processes at scale $\alpha = \frac{1}{T}$ and confidence $\delta$ such that:

$$\log|\mathcal{G}| \leq d\log T + \left(5\sum_{i=1}^{d}\mathsf{VC}(\mathcal{H}_i)\log^2 T\right) + d\log T\log(d/T) + d\log T.$$

The result follows by applying Theorem 2 and taking $\alpha = \delta = \frac{1}{T}$. $\qquad\square$

Note that when $d = 2$ and $\mathcal{H}$ being the class in Example 2, Corollary 2 substantially improves the $O(\sqrt{T})$ regret bound of Bhatt & Kim (2021). Moreover, Bilodeau et al. (2021) derive an $O\left(\frac{\mathsf{VC}(\mathcal{H})\log^2 T}{T}\right)$ *risk bound* under Log-loss, which can be converted to an $O(\mathsf{VC}(\mathcal{H})\log^3 T)$ regret bound via the epoch approach of Lazaric & Munos (2009)[4]. This is off by a $\log T$ factor when compared to our regret in Corollary 2. Furthermore, our results hold for general regret $\tilde{r}_T$ not *average* (and *well specified*) regrets $\bar{r}_T$ as in Bhatt & Kim (2021); Bilodeau et al. (2021).

**Remark 3.** *Note that using a similar argument as in the proof of Theorem 4 and the* multi-class *one-inclusion graph algorithm in (Rubinstein et al., 2006), one can establish an $O(\mathsf{P}(\mathcal{H})\log^2(T/\alpha) + \log(T/\alpha)\log(1/\delta))$ stochastic sequential covering bound for any class $\mathcal{H} \subset [0,1]^{\mathcal{X}}$ with Pseudo-dimension $\mathsf{P}(\mathcal{H})$. See Appendix E.*

### 4.2 Tight bounds for classes with finite Star number

In the previous section, we demonstrated that the stochastic sequential covering number of finite VC class is upper bounded w.h.p. by $e^{O(\log^2 T)}$. We now show that if we assume additional structure on the class, we can improve the bound to $e^{O(\log T)}$, matching the naive fixed design lower bound for many non-trivial classes. In Appendix F we show that even for 1-dimensional threshold functions the *realizable* cumulative error is lower bounded by $\Omega(\log T)$, thus arguing that the error pattern counting argument as in Lemma 2 cannot provide a bound better than $e^{O(\log^2 T)}$. To resolve this issue, we introduce the notion of *Star number* that was used originally in Hanneke & Yang (2015) for analyzing the sample complexity of active learning; however, we use it here in a completely different context. For

---

[4]We believe the regret bound as in the second part of Bilodeau et al. (2021, Theorem 7) missed a $\log n$ factor.

any binary valued class $\mathcal{H}$ and $\mathbf{x}^d \in \mathcal{X}^d$, we say $\mathcal{H}$ Star-shatters $\mathbf{x}^d$ if there exist $h, h_1, \cdots, h_d \in \mathcal{H}$ such that for all $i, j \in [d]$ with $j \neq i$ we have:

$$h(\mathbf{x}_i) \neq h_i(\mathbf{x}_i) \text{ but } h(\mathbf{x}_j) = h_i(\mathbf{x}_j),$$

i.e., a sequence $\mathbf{x}^d$ is Star-shattered by $\mathcal{H}$ if there exists a function $h \in \mathcal{H}$ such that any labeling on $\mathbf{x}^d$, which differs by one position from the realization of $h$, is also realizable by some function $h_i \in \mathcal{H}$. Such a sequence $\mathbf{x}^d$ is called a *star* set of $\mathcal{H}$. The Star number $\mathbf{Star}(\mathcal{H})$ of $\mathcal{H}$ is defined to be the maximum number $d$ such that there exists $\mathbf{x}^d$ that is Star-shattered by $\mathcal{H}$. Clearly, we have $\mathbf{Star}(\mathcal{H}) \geq \mathsf{VC}(\mathcal{H})$ for all $\mathcal{H}$.

We now introduce a new notion of *certification*, which is the key for our following arguments. For any sequence $\mathbf{x}^t$ and $h \in \mathcal{H}$, we say $\mathbf{x}^{t-1}$ certifies $\mathbf{x}_t$ under $h$ if:

$$\forall f \in \mathcal{H}, \text{ if } \forall i \in [t-1], \ f(\mathbf{x}_i) = h(\mathbf{x}_i) \text{ then } f(\mathbf{x}_t) = h(\mathbf{x}_t).$$

We have the following property of finite Star number classes w.r.t. certification:

**Lemma 3.** *If $\mathcal{H}$ has Star number upper bounded by $s$, then for any $\boldsymbol{x}^t \in \mathcal{X}^t$ and $h \in \mathcal{H}$ we have:*

$$\mathrm{Pr}_\sigma \left[ \{\boldsymbol{x}_{\sigma(1)}, \cdots, \boldsymbol{x}_{\sigma(t-1)}\} \text{ certifies } \boldsymbol{x}_{\sigma(t)} \text{ under } h \} \right] \geq 1 - \frac{s}{t},$$

*where $\sigma$ is the uniform random permutation over $[t]$.*

*Proof.* We only need to show that there are at most $s$ points in $\mathbf{x}^t$ that can not be certified by the others under $h$. Suppose otherwise, that we have $s+1$ such points. Consider the realization of $h$ on these points. By definition of certification, we can find functions $h_1, \cdots, h_{s+1}$ as in the definition of Star-shattering. This contradicts the fact that the Star number is upper bounded by $s$. $\square$

We now prove a high probability bound on the number of non-certified positions for a finite Star number class, which is similar to Lemma 1.

**Lemma 4.** *Let $\mathcal{H} \subset \{0,1\}^\mathcal{X}$ be a class with a finite Star number and $T \geq e^5$. Then, with probability $\geq 1 - \delta$ over $\boldsymbol{x}^T$ (sampled from some i.i.d. distribution over $\mathcal{X}^T$) for all $h \in \mathcal{H}$:*

$$\sum_{t=1}^{T} 1\{\boldsymbol{x}^{t-1} \text{ does not certify } \boldsymbol{x}_t \text{ under } h\} \leq \mathsf{VC}(\mathcal{H}) \log T + 4\mathbf{Star}(\mathcal{H}) \log T + \log(1/\delta).$$

*Proof.* Note that the event $\{\mathbf{x}^{t-1} \text{ does not certify } \mathbf{x}_t \text{ under } h\}$ can be viewed as the event $\{\Phi \text{ makes an error at step } t\}$ as in Lemma 1 (since certification is permutation invariant). By Lemma 3 and Lemma 1 with $C = \mathbf{Star}(\mathcal{H})$, we have, for all $h \in \mathcal{H}$ and $\mathbf{x}^T \in \mathcal{X}^T$ w.p. $\geq 1 - \delta$ over uniform random permutation $\sigma$ on $[T]$:

$$\sum_{t=1}^{T} 1\{\mathbf{x}^{\sigma(t-1)} \text{ does not certify } \mathbf{x}_{\sigma(t)} \text{ under } h\} \leq 4\mathbf{Star}(\mathcal{H}) \log T + \log(1/\delta).$$

The result then follows from a similar path as in the proof of Theorem 4 $\square$

Lemma 4 allows us to construct the sequential covering set explicitly without relying on error pattern counting as shown next.

**Theorem 5.** *Let $\mathcal{H}$ be a binary valued class with finite Star number. Then, there exists a stochastic sequential covering set $\mathcal{G}$ of $\mathcal{H}$ w.r.t. the class of all i.i.d. distributions over $\mathcal{X}^T$ at scale $\alpha = 0$ and confidence $\delta$ such that for $T \geq e^5$:*

$$\log |\mathcal{G}| \leq 5\mathbf{Star}(\mathcal{H}) \log T + \log(1/\delta).$$

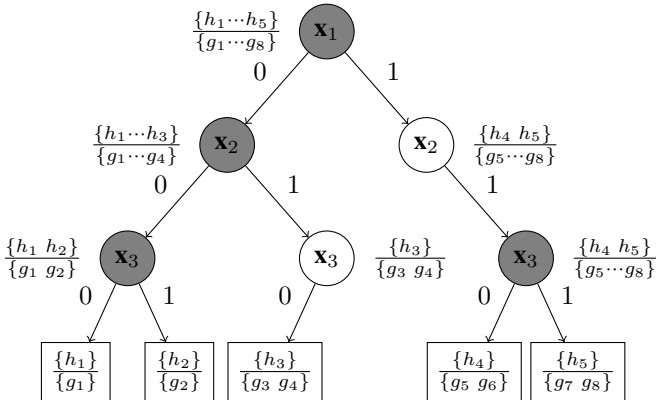

| | $h_1$ | $h_2$ | $h_3$ | $h_4$ | $h_5$ |
|---|---|---|---|---|---|
| $\mathbf{x}_1$ | 0 | 0 | 0 | 1 | 1 |
| $\mathbf{x}_2$ | 0 | 0 | 1 | 1 | 1 |
| $\mathbf{x}_3$ | 0 | 1 | 0 | 0 | 1 |

Figure 1: Realization tree of $\mathcal{H}$ defined in left table and partition of $\mathcal{G}$

*Sketch of Proof.* We only sketch the main idea here (illustrated in Figure 1) and refer to Appendix B for full proof. We will construct a set $\mathcal{G}$ of sequential functions with index set $K$. We select $K$ to be *fixed* with $|K| = 2^M$, where $M = \lceil 5\mathbf{Star}(\mathcal{H}) + \log(1/\delta) \rceil$. We will assign sequentially the value $g_k(\mathbf{x}^t)$ for each $k \in K$ after observing the samples $\mathbf{x}^t$. To do so, we maintain a binary tree called the *realization* tree, where at each time after receiving sample $\mathbf{x}_t$, we expand the leaves of the tree with one or two children depending whether the functions in $\mathcal{H}$ consistent with the leaf agree on $\mathbf{x}_t$ or not. Meanwhile, we also associate with each node in the realization tree a *subset* of $K$ (i.e., a subset of $\mathcal{G}$) in a top-down fashion. Initially, we assign $K$ to the root. Any time we encounter a node with one child, we pass the set associated with that node to its child, else we split the set into two disjoint subsets of *equal* size and pass them to its two children, respectively. The process is said to have failed, if a node with two children has the size of associated set $\leq 1$. If the procedure does not fail until time $T$, for any $k \in K$, there will be exactly one path (from root to leaf) in the realization tree with nodes that have the associated sets containing $k$. The value of $g_k(\mathbf{x}^t)$ is determined by tracing the path. Clearly, if the procedure does not fail, the set $\{g_k : k \in K\}$ sequentially covers $\mathcal{H}$. The key observation is that if we choose $M$ to be large enough, the procedure does not fail with high probability due to Lemma 4 (since only non-certified positions incur splits on associated sets that reduce the size of the sets for children by $1/2$). $\qquad\square$

**Example 5.** *We illustrate the construction of the realization tree in this example. We set $\mathcal{H} = \{h_1, \cdots, h_5\}$, as shown in the table of Figure 1 with sample $\boldsymbol{x}_1, \boldsymbol{x}_2, \boldsymbol{x}_3$. The realization tree is shown in Figure 1, where each function $h \in \mathcal{H}$ is consistent with some path of the tree, and each path has some function $h \in \mathcal{H}$ consistent with it. We assign a subset of $\mathcal{G}$ for each node in the tree denoted as $\{\cdot\}$. Observe that if a node has only one child then the child has the same assigned set as the parent, else we assign an arbitrary partition of the parent set with* equal *sizes to its two children. The final partitions of the set $\mathcal{G} = \{g_1, \cdots, g_8\}$ are in the leaf nodes of the tree. In the figure, binary nodes (i.e., nodes with two children) are in gray color. The maximum number of binary nodes in any path is 3, by selecting $|\mathcal{G}| \geq 2^3 = 8$, which guarantees that the assigning procedure does not fail until the leaf. Each $g_k$ is associated with a unique path from root to the leaf with (the only) assigned sets on the nodes that contain $g_k$. The values of $g_k$ are defined to be the labels of out edges along the path in the obvious way. One can verify that $g_1$ covers $h_1$, $g_2$ covers $h_2$, $g_3, g_4$ covers $h_3$, $g_5, g_6$ covers $h_4$, and $g_7, g_8$ covers $h_5$. Generally, by Lemma 4 the number of binary nodes in any path is of order $O(\log T)$ with high probability (i.e., setting $|\mathcal{G}| = 2^{O(\log T)}$ ensures the process success w.h.p.).*

**Corollary 3.** *Let $\mathcal{H} \subset [0,1]^{\mathcal{X}}$ be the F-composition class as in Example 2 with $\mathcal{H}_1 \subset \{0,1\}^{\mathcal{X}}$ being a class of finite Star number, $\mathcal{P}$ being the class of all i.i.d. distributions over $\mathcal{X}^T$. If $\ell$ is the Log-loss, then:*

$$\tilde{r}_T(\mathcal{H}, \mathcal{P}) \leq O(\mathbf{Star}(\mathcal{H}_1) \log T).$$

*Proof.* By Proposition 2 and Theorem 5, $\mathcal{H}$ admits a stochastic sequential covering set $\mathcal{G}$ at scale $\alpha$ and confidence $\delta$ such that $\log |\mathcal{G}| \leq 2\log(1/\alpha) + 5\mathbf{Star}(\mathcal{H}_1)\log T + \log(1/\delta)$. Taking $\alpha = \delta = \frac{1}{T}$ and applying Theorem 2, the result follows. $\qquad\square$

Note that a natural class that has finite star number is the threshold functions $\mathcal{H} = \{1\{x \geq a\} : x, a \in [0,1]\}$, which has Star number 2. Corollary 3 implies the regret under Log-loss is upper bounded by $O(\log T)$. We refer to (Hanneke & Yang, 2015) for more non-trivial examples. We note also that the $O(\log T)$ regret bound is not likely to be established by the epoch based approach (which Lazaric & Munos (2009); Bhatt & Kim (2021); Bilodeau et al. (2021) have used to establish their regret bounds), since the epochs will inevitably introduce an additional $\log T$ factor.

We observe that being finite Star number is not a necessary condition to achieve a $e^{O(\log T)}$ cover. To see this, consider the class that labels only one sample being 1 and zeros otherwise, which admits a (exact) sequential covering of size $T = e^{\log T}$ but has infinite Star number.

## 4.3 Tight bounds with finite Star-Littlestone dimension

In this section, we introduce a new complexity measure that we call *Star-Littlestone* dimension [5]. The main purpose of this measure is to incorporate the Star number and Littlestone dimension that goes beyond simple finite Star number, and allows us to expand the class of $\mathcal{H}$ with $e^{O(\log T)}$ cover.

**Definition 2** (Star-Littlestone dimension). *Let $\{0,1\}_*^d$ be the set of binary sequences of length less than or equal to $d$. For any numbers $d$ and $s$, we say a binary tree $\tau : \{0,1\}_*^d \to \mathcal{X}$ is Star-Littlestone shattered by $\mathcal{H}$ at star scale $s$ if for any path $\epsilon^d \in \{0,1\}_*^d$ we have $\mathbf{Star}(\mathcal{H}_{\epsilon^d}) > s$, where $\mathcal{H}_{\epsilon^d} = \{h \in \mathcal{H} : \forall t \in [d], h(\tau(\epsilon^{t-1})) = \epsilon_t\}$ and $\mathcal{H} = \cup_{\epsilon^d} \mathcal{H}_{\epsilon^d}$. In words, Star-Littlestone shattering implies that the Star number of the class of hypothesis consistent with any path in $\tau$ has Star number greater than $s$. We define the Star-Littlestone dimension $\mathsf{SL}(s)$ of $\mathcal{H}$ at star scale $s$ to be the maximum number $d$ such that there exists a tree $\tau$ of depth $d$ that is Star-Littlestone shattered at star scale $s$ by $\mathcal{H}$.*

Applying Theorem 5 and the SOA argument as in Ben-David et al. (2009), we establish our next main theorem with a detailed proof in Appendix C.

**Theorem 6.** *Let $\mathcal{H}$ be a binary valued class with Star-Littlestone dimension $\mathsf{SL}(s)$ at star scale $s$. Then, there exists a stochastic sequential covering set $\mathcal{G}$ of $\mathcal{H}$ w.r.t. the class of all i.i.d. distributions over $\mathcal{X}^T$ at scale $\alpha = 0$ and confidence $\delta$ such that:*

$$\log |\mathcal{G}| \leq O(\max\{\mathsf{SL}(s)+1, s\} \log T + \log(1/\delta)).$$

**Example 6.** *In this example, we show a class $\mathcal{H}$ that has both infinite Star number and Littlestone dimension but finite Star-Littlestone dimension. Let $\mathcal{H} = \{h_{[a,b]}(x) = 1\{x \in [a,b]\} : [a,b] \subset [0,1]\}$ be the indicators of intervals. It is easy to see that $\mathcal{H}$ has both infinite Star number and Littlestone dimension. However we can show that it has Star-Littlestone dimension 0 at star scale 4. To see this, consider any point $x \in [0,1]$ and the hypothesis class $\mathcal{H}_1 = \{h \in \mathcal{H} : h(x) = 1\}$. We show that the Star number of $\mathcal{H}_1$ is $\leq 4$. For any 5 points in $[0,1]$, there must be at least 3 points on the same side relative to $x$, the restriction of $\mathcal{H}_1$ on such points is equivalent to threshold functions (either of form $1\{x \geq a\}$ or $1\{x \leq b\}$), thus it cannot Star-shatter these 3 points. This implies that the global sequential covering size of $\mathcal{H}$ is upper bounded by $e^{O(\log T)}$ as in Theorem 6.*

**Example 7.** *Let*

$$\mathcal{H} = \left\{ h_B(\boldsymbol{x}) = 1\{\boldsymbol{x} \in B\} : B = \prod_{i=1}^d [a_i, b_i] \subset \mathbb{R}^d \right\}$$

*be the class of indicators of rectangular cuboids in $\mathbb{R}^d$. Note that $\mathcal{H}$ has infinite Star-Littlestone dimension for any finite star scale when $d \geq 2$ and the VC-dimension of $\mathcal{H}$ is upper bounded by $O(d)$. By Example 4, we have $\mathcal{H}$ can be expressed as a function in terms of indicators of intervals. Applying Proposition 2 and Example 6 we obtain a covering set $\mathcal{G}$ of $\mathcal{H}$ with $\log |\mathcal{G}| \leq O(d \log T + d \log(d/\delta))$. This implies a regret bound of mixable losses (including logarithmic loss) of order $O(d \log T + d \log d)$.*

**Remark 4.** *We leave it as an open problem to determine if the upper bound $e^{O(\log T)}$ can be achieved for any finite VC-dimensional class. Establishing such a result even for the threshold functions $\mathcal{H} = \{h_{\boldsymbol{w}}(\boldsymbol{x}) = 1\{\langle \boldsymbol{w}, \boldsymbol{x} \rangle \geq a\} : \boldsymbol{w}, \boldsymbol{x} \in \mathbb{R}^d, a \in \mathbb{R}\}$ with $d \geq 2$ seems to be a hard task.*

---

[5]This is conceptually similar to the VCL tree as introduced in the recent paper (Bousquet et al., 2021), but they considered a completely different problem using similar idea.

## 5 Real Valued Class with Finite Fat-shattering

We have established tight stochastic sequential covering bound for finite VC-class (and small real valued classes such as finite Pseudo-dimensional classes) in the previous section. We now assume that $\mathcal{H} \subset [0,1]^{\mathcal{X}}$ is a general $[0,1]$-valued function class with bounded fat-shattering number.

We first recall the notion of fat-shattering number, which can be viewed as a scale sensitive VC-dimension. For any class $\mathcal{H} \subset [0,1]^{\mathcal{X}}$, we say $\mathcal{H}$ $\alpha$-fat shatters $\mathbf{x}^d \in \mathcal{X}^d$ if there exists $s^d \in [0,1]^d$ such that for all $I \subset [d]$ there exists $h \in \mathcal{H}$ such that for all $t \in [d]$: (i) If $t \in I$, then $h(\mathbf{x}_t) \geq s_t + \alpha$; and (ii) If $t \notin I$, then $h(\mathbf{x}_t) \leq s_t - \alpha$. Then, the fat shattering number of $\mathcal{H}$ at scale $\alpha$ is defined to be the maximum number $d := d(\alpha)$ such that there exists $\mathbf{x}^d \in \mathcal{X}^d$ with $\mathcal{H}$ $\alpha$-fat shatters $\mathbf{x}^d$.

We now state our main result for this section.

**Theorem 7.** *Let $\mathcal{H}$ be a class of functions $\mathcal{X} \to [0,1]$ with the $\alpha$-fat shattering number $d(\alpha)$. Then there exists a stochastic global sequential covering set $\mathcal{G}$ of $\mathcal{H}$ w.r.t. the class of all i.i.d. distributions over $\mathcal{X}^T$ at scale $\alpha$ and confidence $\delta$ such that:*

$$\log |\mathcal{G}| \leq O(d(\alpha/32)(\log T \log(4/\alpha))^4 + (\log^2 T + \log T \log(4/\alpha)) \log(\log T/\delta)),$$

*where $O$ hides absolute constant which is independent of $\alpha$, $T$, and $\delta$.*

**Remark 5.** *It is easy to show that for any class $\mathcal{H}$ with $\alpha$-fat shattering number of order $d(\alpha)$, the stochastic sequential covering number must be lower bounded by $d(\alpha)$. This can be seen by considering the sample $\mathbf{x}^{d(\alpha)}$ that is $\alpha$-fat shattered by $\mathcal{H}$, since one cannot $\alpha$-cover any two distinct functions that witness the $\alpha$-fat shattering on $\mathbf{x}^{d(\alpha)}$ using a single (sequential) function. However, our logarithmic dependency may not be tight, and we leave it as an open problem to obtain the optimal dependency.*

We first introduce the notion of a local $\alpha$-covering. We say that a class $\mathcal{F}$ locally $\alpha$-covers $\mathcal{H}$ at $\mathbf{x}^T \in \mathcal{X}^T$ if for all $h \in \mathcal{H}$ there exists $f \in \mathcal{F}$ such that:

$$\forall t \in [T], \ |h(\mathbf{x}_t) - f(\mathbf{x}_t)| \leq \alpha.$$

Here, we also assume that $\mathcal{F} \subset \mathcal{H}$ (we can always convert $\alpha$-covering set $\mathcal{F}$ of $\mathcal{H}$ to a $2\alpha$-covering set $\tilde{\mathcal{F}} \subset \mathcal{H}$ such that $|\tilde{\mathcal{F}}| \leq |\mathcal{F}|$).

The following lemma upper bounds the local $\alpha$-covering size w.r.t. the $\alpha$-fat shattering number of $\mathcal{H}$, which is due to Alon et al. (1997).

**Lemma 5.** *Suppose the $\alpha$ fat-shattering number of $\mathcal{H}$ is $d(\alpha)$. Then for all $\mathbf{x}^T \in \mathcal{X}^T$, there exists $\mathcal{F}$ (which depends on $\mathbf{x}^T$) that locally $\alpha$-covers $\mathcal{H}$ at $\mathbf{x}^T$ such that:*

$$|\mathcal{F}| \leq 2 \left( T \left( \frac{2}{\alpha} + 1 \right)^2 \right)^{\left\lceil d(\alpha/4) \log \left( \frac{2eT}{\alpha d(\alpha/4)} \right) \right\rceil} \leq 2^{d(\alpha/4)(\log^2 T + 2\log^2(1/\alpha) + O(1))}.$$

Our proof of Theorem 7 is based on the following key lemma (which is an application of the classical symmetrization argument), and an epoch approach similar to Lazaric & Munos (2009).

**Lemma 6.** *Let $\mathcal{H} \subset [0,1]^{\mathcal{X}}$ be a class with $\alpha$-fat shattering number $d(\alpha)$. Let $S_1, S_2$ be two i.i.d. samples from the same distribution over $\mathcal{X}$, both of size $k$. For any $S_i$ with $i \in \{1, 2\}$, we define a distance for all $h_1, h_2 \in \mathcal{H}$ as:*

$$d_{S_i}^{\alpha}(h_1, h_2) = \sum_{s \in S_i} \mathbf{1}\{|h_1(s) - h_2(s)| \geq \alpha\}.$$

*Then*

$$\Pr_{S_1, S_2} \left[ \exists h_1, h_2 \in \mathcal{H} \ s.t. \ d_{S_1}^{\alpha}(h_1, h_2) = 0 \ and \ d_{S_2}^{4\alpha}(h_1, h_2) \geq r \right] \leq 2^{\tilde{O}(d(\alpha/8)) - r},$$

*where $\tilde{O}(d(\alpha/8)) = 2d(\alpha/8)(\log^2 k + 2\log^2(1/\alpha) + O(1))$.*

*Proof.* We use a symmetrization argument. We denote by $A$ the event that $\exists h_1, h_2 \in \mathcal{H}$ such that $d_{S_1}^\alpha(h_1, h_2) = 0$ but $d_{S_2}^{4\alpha}(h_1, h_2) \geq r$. Let $\sigma$ be a random permutation that switches the $i$th positions of $S_1, S_2$ w.p. $\frac{1}{2}$ and independently for different $i \in [k]$. By symmetries, it is sufficient to fix $S_1, S_2$ and upper bound $\Pr_\sigma[A[\sigma(S_1, S_2)]]$. By Lemma 5, we know that there exists a set $\mathcal{F}$ that $\alpha/2$-covers $\mathcal{H}$ on $S_1 \cup S_2$ with:

$$|\mathcal{F}| \leq 2^{d(\alpha/8)(\log^2 k + 2\log^2(1/\alpha) + O(1))}.$$

If the event $A$ happens, then there exist $f_1, f_2 \in \mathcal{F}$ such that (using property of covering):

$$d_{S_1}^{2\alpha}(f_1, f_2) = 0 \text{ but } d_{S_2}^{3\alpha}(f_1, f_2) \geq r.$$

Clearly, in order for $A$ to happen, any position $s \in S_2$ such that $|f_1(s) - f_2(s)| \geq 3\alpha$ must not be switched to $S_1$ under $\sigma$, which happens with probability upper bounded by $2^{-r}$. Applying union bound over all pairs of $\mathcal{F}$, we have

$$\Pr_{S_1, S_2}[A] \leq 2^{2d(\alpha/8)(\log^2 k + 2\log^2(1/\alpha) + O(1)) - r}$$

which completes the proof. $\qquad\square$

*Proof of Theorem 7.* We partition the time horizon into epochs, where each epoch $s$ ranges from time step $2^{s-1}, \cdots, 2^s - 1$. For each epoch $s$, we will construct a covering set $\mathcal{G}_s$. The global covering set $\mathcal{G}$ will be constructed by considering all the combinations of functions in $\mathcal{G}_s$ with $s \in \{1, \cdots, \lceil \log T \rceil\}$.

For any epoch $s$, we construct $\mathcal{G}_s$ as follows. Let $\mathcal{F} \subset \mathcal{H}$ be the local $\alpha$-covering set on the samples $\mathbf{x}_1, \cdots, \mathbf{x}_{2^{s-1}-1}$. By Lemma 5, we have

$$|\mathcal{F}| \leq 2^{d(\alpha/4)(s^2 + 2\log^2(1/\alpha) + O(1))}.$$

Let

$$r_s = 2d(\alpha/8)(s^2 + 2\log^2(1/\alpha) + O(1)) + \log(\log T/\delta).$$

By Lemma 6 w.p. $\geq 1 - \frac{\delta}{\log T}$ for any $h \in \mathcal{H}$ there exists $f \in \mathcal{F}$ such that $f$ $4\alpha$-covers $h$ on samples $\mathbf{x}_{2^{s-1}}, \cdots, \mathbf{x}_{2^s-1}$ except $r_s$ positions (the $f \in \mathcal{F}$ that $\alpha$-covers $h$ on $\mathbf{x}^{2^{s-1}-1}$ is the desired function since $\mathcal{F}$ is a local $\alpha$-covering). Let $J$ be a discretization of interval $[0, 1]$ such that for any $a \in [0, 1]$, there exists $b \in J$ so that $|a - b| \leq 4\alpha$. We have $|J| \leq \lceil \frac{1}{8\alpha} \rceil$. Now, for any $I \subset \{2^{s-1}, \cdots, 2^s - 1\}$ with $|I| \leq r_s$, $\{k_i\}_{i \in I} \in J^{|I|}$ and $f \in \mathcal{F}$, we construct a function $f_{I,k^{|I|}}$ as follows:

1. If $t \in I$, we set $f_{I,k^{|I|}}(\mathbf{x}_t) = k_t$;

2. If $t \notin I$, we set $f_{I,k^{|I|}}(\mathbf{x}_t) = f(\mathbf{x}_t)$.

The class $\mathcal{G}_s$ is defined as the class of all such $f_{I,k^{|I|}}$. By definition of $r_s$ and by Lemma 6, we have w.p. $\geq 1 - \frac{\delta}{\log T}$, for all $h \in \mathcal{H}$ there exists $g \in \mathcal{G}_s$ such that for all $t \in \{2^{s-1}, \cdots, 2^s - 1\}$ we have:

$$|g(\mathbf{x}_t) - h(\mathbf{x}_t)| \leq 4\alpha.$$

We observe that:

$$|\mathcal{G}_s| \leq |\mathcal{F}| \cdot (2^s |K|)^{r_s + 1} \leq 2^{O(d(\alpha/8)((s\log(1/\alpha))^3) + (s + \log(1/\alpha))\log(\log T/\delta))}.$$

We now construct the global covering set $\mathcal{G}$ as follows. For any index $(j_1, \cdots, j_{\lceil \log T \rceil})$ with $j_s \in [|\mathcal{G}_s|]$, we define a function $g$ such that it uses the $j_s$ function in $\mathcal{G}_s$ to make prediction during epoch $s$. By union bound on the epochs, we have w.p. $\geq 1 - \delta$ for any $h \in \mathcal{H}$, there exists $g$ such that:

$$\forall t \in [T], |h(\mathbf{x}_t) - g(\mathbf{x}^t)| \leq 4\alpha.$$

This implies that $\mathcal{G}$ is a $4\alpha$ global sequential covering set of $\mathcal{H}$. Thus

$$|\mathcal{G}| = \prod_{s=1}^{\lceil \log T \rceil} |\mathcal{G}_s| \leq 2^{O(d(\alpha/8)(\log T \log(1/\alpha))^4 + (\log^2 T + \log T \log(1/\alpha))\log(\log T/\delta))}.$$

The result follows by taking $\alpha$ in the above expression to be $\alpha/4$. $\qquad\square$

We complete this section with two results regarding the expected worst case minimax regret.

**Corollary 4.** *Let $\mathcal{H}$ be a [0,1]-valued class with $\alpha$-fat shattering number of order $\alpha^{-l}$ for some $l \geq 0$, and $\mathcal{P}$ be a class of all $i.i.d.$ distributions over $\mathcal{X}^T$. If $\ell(\cdot, y)$ is convex, L-Lipschitz and bounded by 1 for all $y \in \mathcal{Y}$, then:*

$$\tilde{r}_T(\mathcal{H}, \mathcal{P}) \leq \tilde{O}((LT)^{(l+1)/(l+2)})$$

*where $\tilde{O}$ hides a poly-log factor.*

*Proof.* Apply Theorem 7 to Theorem 1 to find $\tilde{r}_T(\mathcal{H}, \mathcal{P}) \leq \inf_{0 \leq \alpha \leq 1} \left\{ \alpha LT + \tilde{O}\left(\sqrt{T\alpha^{-l}}\right) \right\}$ and taking $\alpha = (LT)^{-1/(l+2)}$ finishes the proof. $\square$

Note that Block et al. (2022, Theorem 3) demonstrated that for *known i.i.d.* processes one can achieve an $\tilde{O}(T^{(l-1)/l})$ regret bound (in fact they establish the result for the *smooth adversary* processes). However, extending such an chaining based argument to our *unknown i.i.d.* processes as in Corollary 4 seems to be an non-trivial task, since for unknown $i.i.d.$ processes one cannot express the expected worst case regret in the iterated minimax formulation as in Rakhlin et al. (2011) (see Remark 1). We leave it as an open problem to determine if the bound in Corollary 4 is tight or not for the unknown $i.i.d.$ processes.

**Corollary 5.** *Let $\mathcal{H}$ be a [0,1]-valued class with $\alpha$-fat shattering number of order $\alpha^{-l}$ with $l \geq 0$, and $\mathcal{P}$ be the class of all $i.i.d.$ distributions over $\mathcal{X}^T$. If $\ell$ is Log-loss, then $\tilde{r}_T(\mathcal{H}, \mathcal{P}) \leq \tilde{O}(T^{l/l+1})$.*

*Proof.* Applying Theorem 7 to Theorem 2, we have $\tilde{r}_T(\mathcal{H}, \mathcal{P}) \leq \inf_{0 \leq \alpha \leq 1} \left\{ 2\alpha T + \tilde{O}(\alpha^{-l}) \right\}$, and taking $\alpha = T^{-1/(l+1)}$ completes the proof. $\square$

We can show that the regret bound in Corollary 5 is actually *tight* upto poly-logarithmic factors for *general* classes of $\alpha$-fat shattering number of order $\alpha^{-l}$ (with $l \geq 1$), see Proposition 3 in Section 6. However, it is known by Bilodeau et al. (2020); Wu et al. (2022b) that this bound is not tight for *all* classes even for the adversary case. Comparing to the results in Bilodeau et al. (2021), Corollary 5 shows that the regret behaviour under Log-loss for $\tilde{r}_T$ is closer to the adversary case as in Wu et al. (2022b) instead of the (well-specified) average case as in Bilodeau et al. (2021).

## 6 Lower Bounds For Regret

We now provide a general approach for lower bounding the regret $\tilde{r}(\mathcal{H}, \mathcal{P})$ using the fixed design regret defined in (2) and analyzed in Wu et al. (2022b) as well as Shamir (2020); Shamir & Szpankowski (2021); Jacquet et al. (2021). We will assume throughout this section that $\mathcal{H} \subset [0,1]^{\mathcal{X}}$ is a general real valued function class and $\mathcal{P}$ is the class of all $i.i.d.$ processes over $\mathcal{X}^T$. We first introduce the following well known tail bound for the coupon collector problem, see e.g. (Doerr, 2020, Theorem 1.9.2).

**Lemma 7.** *Let $X_1, X_2, \cdots$ be $i.i.d.$ samples from the uniform distribution over $[T]$, and $\rho$ be the first time such that $[T] \subset X_1^\rho$. Then for any $c \geq 0$ we have $\Pr[\rho \geq T \log T + cT] \leq e^{-c}$.*

For any function $\Phi$ that maps sequences from $\mathcal{X}^*$ to $\mathbb{R}$, we say $\Phi$ is monotone if for any $\mathbf{x}^T \subset \mathbf{z}^{T_1}$ we have $\Phi(\mathbf{x}^T) \leq \Phi(\mathbf{z}^{T_1})$, where $\mathbf{x}^T \subset \mathbf{z}^{T_1}$ means that for any $\mathbf{s} \in \mathcal{X}$, the number of $\mathbf{s}$ appearances in $\mathbf{x}^T$ is no more than the number of appearances of $\mathbf{s}$ in $\mathbf{z}^{T_1}$. We also assume a regularity condition for the loss $\ell$ such that for all $\hat{y}_1, \hat{y}_2 \in \mathcal{Y}$ there exists $y \in \mathcal{Y}$ with $\ell(\hat{y}_1, y) \geq \ell(\hat{y}_2, y)$. We also recall that $r_T^*(\mathcal{H}) = \sup_{\mathbf{x}^T} r^*(\mathcal{H}|\mathbf{x}^T)$.

**Theorem 8.** *Let $\mathcal{H}$ be any $[0,1]$-valued class. If the fixed design regret $r_T^*(\mathcal{H} \mid \mathbf{x}^T)$, as defined in (2), is monotone over $\mathbf{x}^T$ and $\ell$ satisfies the above regularity condition, then:*

$$\tilde{r}_T(\mathcal{H}, \mathcal{P}) \geq (1 - O(1/\log T)) r_{\kappa^{-1}(T)}^*(\mathcal{H}) \geq (1 - O(1/\log T)) r_{(T/\log T)}^*(\mathcal{H}),$$

*where $\mathcal{P}$ is the class of all $i.i.d.$ distributions over $\mathcal{X}^T$ and $\kappa(T) = T \log T + T \log \log T$.*

*Proof.* Let $\tilde{\mathbf{x}}^T$ be the feature that achieves the maximum of $r_T^*(\mathcal{H} \mid \tilde{\mathbf{x}}^T)$ (i.e., $r_T^*(\mathcal{H})$). We define the distribution $\nu$ to be the uniform distribution over $\{\tilde{\mathbf{x}}_1, \cdots, \tilde{\mathbf{x}}_T\}$ (with possibly repeated elements). Let $T_1 = T \log T + T \log \log T$. We have

$$\tilde{r}_{T_1}(\mathcal{H}, \mathcal{P}) = \inf_{\phi^{T_1}} \sup_{\mu \in \mathcal{P}} \mathbb{E}_{\mathbf{x}^{T_1} \sim \mu} \left[ \sup_{y^{T_1}} \left( \sum_{t=1}^{T_1} \ell(\hat{y}_t, y_t) - \inf_{h \in \mathcal{H}} \sum_{t=1}^{T_1} \ell(h(\mathbf{x}_t), y_t) \right) \right] \tag{5}$$

$$\geq \inf_{\phi^{T_1}} \mathbb{E}_{\mathbf{x}^{T_1} \sim \nu^{T_1}} \left[ \sup_{y^{T_1}} \left( \sum_{t=1}^{T_1} \ell(\hat{y}_t, y_t) - \inf_{h \in \mathcal{H}} \sum_{t=1}^{T_1} \ell(h(\mathbf{x}_t), y_t) \right) \right] \tag{6}$$

$$\overset{(a)}{\geq} \inf_{\phi^{T_1}} \Pr[\tilde{\mathbf{x}}^T \subset \mathbf{x}^{T_1}] \cdot \mathbb{E} \left[ \sup_{y^{T_1}} \left( \sum_{t=1}^{T_1} \ell(\hat{y}_t, y_t) - \inf_{h \in \mathcal{H}} \sum_{t=1}^{T_1} \ell(h(\mathbf{x}_t), y_t) \right) \mid \tilde{\mathbf{x}}^T \subset \mathbf{x}^{T_1} \right] \tag{7}$$

$$\overset{(b)}{\geq} \Pr[\tilde{\mathbf{x}}^T \subset \mathbf{x}^{T_1}] \cdot \mathbb{E} \left[ \inf_{\phi^{T_1}} \sup_{y^{T_1}} \left( \sum_{t=1}^{T_1} \ell(\hat{y}_t, y_t) - \inf_{h \in \mathcal{H}} \sum_{t=1}^{T_1} \ell(h(\mathbf{x}_t), y_t) \right) \mid \tilde{x}^T \subset \mathbf{x}^{T_1} \right] \tag{8}$$

$$= \Pr[\tilde{\mathbf{x}}^T \subset \mathbf{x}^{T_1}] \cdot \mathbb{E} \left[ r_{T_1}^*(\mathcal{H} \mid \mathbf{x}^{T_1}) \mid \tilde{\mathbf{x}}^T \subset \mathbf{x}^{T_1} \right] \tag{9}$$

$$\overset{(c)}{\geq} \Pr[\tilde{\mathbf{x}}^T \subset \mathbf{x}^{T_1}] r_T^*(\mathcal{H} \mid \tilde{\mathbf{x}}^T) \overset{(d)}{\geq} \left( 1 - \frac{1}{\log T} \right) r_T^*(\mathcal{H}), \tag{10}$$

where $(a)$ follows by conditioning on the event $\{\tilde{\mathbf{x}}^T \subset \mathbf{x}^{T_1}\}$ and observing that the regret is positive for all $\mathbf{x}^{T_1}$, $(b)$ follows by $\inf \mathbb{E} \geq \mathbb{E} \inf$, $(c)$ follows from the fact that $r_{T_1}^*(\mathcal{H} \mid \mathbf{x}^{T_1}) \geq r_T^*(\mathcal{H} \mid \tilde{\mathbf{x}}^T)$ which further follows from the monotonicity of $r_T^*(\mathcal{H} \mid \mathbf{x}^T)$, $(d)$ follows by Lemma 7. To complete the proof we have $T = \kappa^{-1}(T_1)$ and notice that $\kappa^{-1}(T_1) \geq \frac{T_1}{\log T_1}$. $\square$

The following lemma shows the monotonicity for Log-loss [6].

**Lemma 8.** *For Log-loss we have $r_{T_1}^*(\mathcal{H} \mid \boldsymbol{x}^{T_1}) \geq r_T^*(\mathcal{H} \mid \tilde{\boldsymbol{x}}^T)$, so long as $\tilde{\boldsymbol{x}}^T \subset \boldsymbol{x}^{T_1}$.*

*Proof.* Note that for any $\mathbf{x}^T$, we have (Jacquet et al., 2021):

$$r_T^*(\mathcal{H} \mid \mathbf{x}^T) = \log \sum_{y^T} \sup_{h \in \mathcal{H}} \prod_{t=1}^{T} h(\mathbf{x}_t)^{y_1} (1 - h(\mathbf{x}_t))^{1-y_t}.$$

Therefore, any permutation over $\mathbf{x}^T$ does not change the value $r_T^*$. Now, suppose $\tilde{\mathbf{x}}^T \subset \mathbf{x}^{T_1}$; we can permute $\mathbf{x}^{T_1}$ so that the first $T$ samples match with $\tilde{\mathbf{x}}^T$. The result follows from the fact that playing more rounds does not decrease the regret. To see this, we let $h \in \mathcal{H}$ to be the hypothesis that achieves minimal accumulated loss in the first $T$ rounds, we then select the label $y_t$ for which $\ell(\hat{y}_t, y_t) \geq \ell(h(\mathbf{x}_t), y_t)$ for the following steps $t > T$, which ensures non-decreasing regret. $\square$

Finally, we apply the above general lower bound to the expected worst case minimax regret.

**Corollary 6.** *Assume $\ell$ is the Log-loss. If $r_T^*(\mathcal{H}) \geq C \log^\alpha T$ then*

$$\tilde{r}_T(\mathcal{H}, \mathcal{P}) \geq C \log^\alpha T - o(\log^\alpha T),$$

*where $\mathcal{P}$ is the class of i.i.d. distributions. If $r_T^*(\mathcal{H}) \geq C T^\alpha$, then*

$$\tilde{r}_T(\mathcal{H}, \mathcal{P}) \geq \frac{C T^\alpha}{\log^\alpha T} - o(T^\alpha / \log^\alpha T).$$

**Remark 6.** *A question arises whether the $\log T$ factor in Corollary 6 can be eliminated. We do not have a complete answer for this question at this point; however, we will show in Appendix D that there exists a class $\mathcal{H}$ such that $\tilde{r}_T(\mathcal{H}, \mathcal{P}) \leq (1 - e^{-1}) r_T^*(\mathcal{H})$, where $\mathcal{P}$ is the class of all i.i.d. processes. Meaning that the reduction as in Corollary 6 will necessarily introduce a factor $< 1$ for polynomial regrets $r_T^*(\mathcal{H})$.*

---

[6]Using the result in Cesa-Bianchi & Lugosi (2006, Theorem 8.1), one can establish similar result for absolute loss.

We refer to (Wu et al., 2022b) for the lower bounds on $r_T^*(\mathcal{H})$ of various classes $\mathcal{H}$ under Log-loss. In particular, the following lower bound is a complement to Corollary 5.

**Proposition 3.** *For any $l \geq 1$, there exists a $[0,1]$-valued class $\mathcal{H}$ with $\alpha$-fat-shattering number of order $O(\alpha^{-l})$ and $\mathcal{P}$ is the class of all i.i.d. distributions over $\mathcal{X}^T$, such that*

$$\tilde{r}_T(\mathcal{H}, \mathcal{P}) \geq \tilde{\Omega}(T^{l/(l+1)}),$$

*under logarithmic loss.*

*Proof.* Let $\mathcal{X} = [T]$; we define $\mathcal{H} = \{h \in [0,1]^{\mathcal{X}} : \sum_{t=1}^T h(t)^l \leq 1\}$. We claim that the $\alpha$-fat shattering number of $\mathcal{H}$ is upper bounded by $\alpha^{-l}$. To see this, we assume there exist $d$ points $\mathbf{x}^d \in [T]$ such that $d > \alpha^{-l}$ and $\mathbf{x}^d$ is $\alpha$-fat shattered by $\mathcal{H}$. By definition of $\alpha$-fat shattering, there exist two functions $h_1, h_2 \in \mathcal{H}$ such that $\forall i \in [d]$, $|h_1(\mathbf{x}_i) - h_2(\mathbf{x}_i)| \geq 2\alpha$. This implies that $\sum_{t=1}^T |h_1(t) - h_2(t)|^l \geq d \cdot (2\alpha)^l > 2^l$, i.e., $||h_1 - h_2||_l > 2$. However, this contradicts the fact that $||h_1 - h_2||_l \leq ||h_1||_l + ||h_2||_l \leq 2$ by the triangle inequality of $L_l$ norm. In conclusion, by Wu et al. (2022b, Theorem 6) we have $r_T^*(\mathcal{H}) \geq \Omega(T^{l/(l+1)})$. Invoking Corollary 6, and the result follows. $\square$

Note that Proposition 3 only shows that the lower bound $\tilde{\Omega}(T^{l/(l+1)})$ holds for certain *hard* classes. We prove in the following proposition a lower bound that holds for *all* classes.

**Proposition 4.** *Let $l \geq 1$, $\mathcal{H}$ be any $[0,1]$-valued class with $\alpha$-fat-shattering number of order $\Omega(\alpha^{-l})$ and $\mathcal{P}$ is the class of all i.i.d. distributions over $\mathcal{X}^T$. Then*

$$\tilde{r}_T(\mathcal{H}, \mathcal{P}) \geq \tilde{\Omega}(T^{(l-1)/l}),$$

*under logarithmic loss.*

*Proof.* Let $\mathbf{x}^T$ be samples that are $\alpha$-fat-shattered by $\mathcal{H}$ and witnessed by $s^T$, where $\alpha \geq \Omega(T^{-1/l})$. We now describe an adversary strategy that achieves the $\Omega(T^{(l-1)/l})$ lower bound for the fixed design regret $r_T^*(\mathcal{H} \mid \mathbf{x}^T)$. To see this, for any $t \in [T]$, if the predictor predicts $\hat{y}_t \geq s_t$, we set $y_t = 0$, else, we set $y_t = 1$. By definition of $\alpha$-fat shattering, there exists $h \in \mathcal{H}$ such that $\forall t \in [T], |h(\mathbf{x}_t) - \hat{y}_t| \geq \alpha$ and $\ell(\hat{y}_t, y_t) \geq \ell(h(\mathbf{x}_t), y_t)$. We assume without loss of generality, $y_t = 1$. By definition of Log-loss, we have:

$$\ell(\hat{y}_t, y_t) - \ell(h(\mathbf{x}_t), y_t) = \log(h(\mathbf{x}_t)/\hat{y}_t) \geq \log((\hat{y}_t + \alpha)/\hat{y}_t) \geq \alpha/2,$$

The last inequality follows by $\log(1 + x) \geq x/(x+1)$. Therefore, we have $r_T^*(\mathcal{H} \mid \mathbf{x}^T) \geq T\alpha/2 \geq \Omega(T^{(l-1)/l})$. The proposition now follows by Corollary 6. $\square$

Note that when $l \geq 2$ the lower bound in Proposition 4 is achieved by Logistic regression (Foster et al., 2018, Example 2). Therefore, the lower bound is not *universally* improvable (this is similar to Corollary 5).

**Example 8** (Well-specified v.s. worst case $y^T$)**.** *In this example we demonstrate that the expected worst case regret $\tilde{r}_T$ can be substantially different than the well-specified average case regret $\bar{r}_T$ as in Bilodeau et al. (2021). This will explain why our Theorem 8 is a necessary technique for establishing lower bonds for $\tilde{r}_T$. To see this, for any $\mathcal{X}$ with $|\mathcal{X}| \geq T$ we define:*

$$\mathcal{H} = \left\{ h_b(\boldsymbol{x}) = \frac{1}{2} + \frac{b(\boldsymbol{x})}{\sqrt{T}} : b \in [-1,1]^{\mathcal{X}} \right\}.$$

*This class admits an $O(1)$ uniform KL-cover at scale $O(1/T)$, and therefore by Bilodeau et al. (2021), the well-specified regret is of order $O(1)$. However, by Wu et al. (2022b, Page 6), the fixed design regret $r_T^*(\mathcal{H}) \geq 2(1/\sqrt{T})T \geq \Omega(\sqrt{T})$. Invoking Corollary 6, this implies an $\tilde{\Omega}(\sqrt{T})$ lower bound for $\tilde{r}_T$. This also demonstrates that the KL-cover (or equivalently the Hellinger cover) as in Bilodeau et al. (2021) cannot capture the behaviour of $\tilde{r}_T$ under Log-loss even with values bounded away from $0$.*

# 7 Conclusion

In this paper we introduced a general minimax regret called the expected worst case minimax regret when the features are generated by a stochastic source. This new minimax regret recovers previously known online minimax regrets in a unified way. We analyzed the regret via a novel concept of stochastic global sequential covering and provide tight bounds on the covering size when the underlying process is $i.i.d..$ A direct generalization is to extend our results to some interesting general random processes. We also expect that the technique of stochastic global sequential covering can be exploited beyond problems studied in this paper, e.g., for contextual bandits with general reference policy sets. From an algorithmic perspective, our prediction rules derived from Theorems 1 and 2 are based on the Exponential Weighted Average algorithm, which is generally not computationally efficient. We believe that investigating computationally efficient algorithms (such as the oracle efficient algorithms in Block et al. (2022)) represents an intriguing direction for future research. We leave these extensions and applications to future work.

# Acknowledgements

This work was partially supported by the NSF Center for Science of Information (CSoI) Grant CCF-0939370, by NSF Grants CCF-2006440, CCF-2007238, CCF- 2211423, and OAC-1908691 and in addition by a Google Research Grant.

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

## A  Proof of Lemma 1

For any $t \in [T]$, we denote $I_t = 1\{\Phi(\mathbf{x}^{\sigma(t)}, h(\{\mathbf{x}^{\sigma(t-1)}\})) \neq h(\mathbf{x}_{\sigma(t)})\}$, where $\sigma$ is a uniform random permutation over $[T]$. For all $t \in [T]$, we define the *reversed* sequence of indicators $I'_t = I_{T-t+1}$. We observe that for all $t \in [T]$, the indicator $I'_t$ only depends on the realizations of $\mathbf{x}_{\sigma(T)}, \mathbf{x}_{\sigma(T-1)}, \cdots, \mathbf{x}_{\sigma(T-t+1)}$ since $\Phi$ is permutation invariant on $\mathbf{x}_{\sigma(1)}, \cdots, \mathbf{x}_{\sigma(T-t)}$. Therefore,

$$\mathbb{E}[I'_t \mid I'_1, \cdots, I'_{t-1}] = \mathbb{E}[I'_t \mid \mathbf{x}_{\sigma(T)}, \cdots, \mathbf{x}_{\sigma(T-t+2)}] \leq \min\left\{\frac{C}{T-t+1}, 1\right\},$$

where the last inequality follows from the assumption of $\Phi$ and noticing that conditioning on $\mathbf{x}^{\sigma(T)}_{\sigma(T-t+2)}$ the permutation $\sigma$ restricted on $\mathbf{x}^T \backslash \{\mathbf{x}_{\sigma(T)}, \cdots, \mathbf{x}_{\sigma(T-t+2)}\}$ is also a uniform random permutation. For any realization $I'_1, \cdots, I'_{t-1}$, we define $I''_t = I'_t - \mathbb{E}[I'_t \mid I'_1, \cdots, I'_{t-1}]$. We now observe that the indicators $I''_t$ are (Doob) martingale differences, i.e., we have $\forall t \in [T]$:

$$\mathbb{E}[I''_t \mid I''_1, \cdots, I''_{t-1}] = 0.$$

By the Bernstein inequality for martingales (Cesa-Bianchi & Lugosi, 2006, Lemma A.8), we find:

$$\Pr\left[\sum_{t=1}^{T} I''_t > k \text{ and } \Sigma^2 \leq v\right] \leq e^{-\frac{k^2}{2(v+k/3)}},$$

where

$$\Sigma^2 = \sum_{t=1}^{T} \mathbb{E}[I''^2_t \mid I''_1, \cdots, I''_{t-1}].$$

We now observe that conditioning on $I'_1, \cdots, I'_{t-1}$, the indicator $I'_t$ is a Bernoulli random variable with parameter $p_t \leq \min\left\{\frac{C}{T-t+1}, 1\right\}$. This implies that if $I'_t = 1$ then $I''_t \leq 1$ and if $I'_t = 0$ then $|I''_t| \leq p_t$. Using elementary algebra, we have with probability 1 that

$$\sum_{t=1}^{T} \mathbb{E}[I''^2_t \mid I'_1, \cdots, I'_{t-1}] \leq \sum_{t=1}^{T} p_t + (1-p_t)p_t^2 \leq C \log T + 3C.$$

Plugging it into the Bernstein inequality with $k = 2(C \log T + 3C) + \log(1/\delta))$ and $v = C \log T + 3C$, with probability $\geq 1 - \delta$, we have:

$$\sum_{t=1}^{T} I_t = \sum_{t=1}^{T} I'_t \leq \sum_{t=1}^{T} I''_t + \mathbb{E}[I'_t \mid I'_1, \cdots, I'_{t-1}]$$

$$\leq k + \sum_{t=1}^{T} \min\left\{\frac{C}{T-t+1}, 1\right\} \leq 3C \log T + 5C + \log(1/\delta).$$

Here, we used the following elementary inequality:

$$\forall a, b \geq 0, \ \frac{(2a+b)^2}{2(a+(2a+b)/3)} \geq b.$$

The Lemma now follows from the fact that $C \log T \geq 5C$ when $T \geq e^5$ and $C \geq 1$.

## B  Proof of Theorem 5

We will construct a covering set $\mathcal{G}$ directly without relying on the error pattern counting as in Lemma 2. This is the key to removing the extra $\log T$ factor. We will introduce a set $K$ to index the functions in $\mathcal{G}$, we assume that $K$ is fixed and $|K| = 2^M$ for some $M$ to be chosen later. For any $k \in K$, we will construct a *sequential function* $g_k$ as follows:

Let $\mathbf{x}^T$ be a realization of the sample from an *i.i.d.* source. The realization tree $\mathcal{T}$ of $\mathcal{H}$ on $\mathbf{x}^T$ is a leveled binary tree of depth $T+1$, with each node at level $t$ being labeled $\mathbf{x}_t$ (where level 1 has only the root $v_1$), each left edge being labeled 0 and each right edge being labeled 1, such that any node $v_t \in \mathcal{T}$ at level $t$ has left (respectively right) child if and only if there exist $h \in \mathcal{H}$ such that $h(\mathbf{x}_t) = 0$ (respectively $h(\mathbf{x}_t) = 1$) and $h(\mathbf{x}_i) = L(v_i \to v_{i+1})$ for all $i \leq t-1$, where $v_1 \to v_2 \to \cdots \to v_t = v$ is the path from root $v_1$ to $v$ and $L$ is the edge label function. Note that different realizations of $\mathbf{x}^T$ will result in different realization trees.

We now assign values of the functions $g_k$ with $k \in K$ using the following procedure. For any node $v$ in the realization tree $\mathcal{T}$, we will associate a set $\mathcal{K}(v) \subset K$ using the following rule (starting from root):

1. If $v$ is the root, then $\mathcal{K}(v) = K$;

2. If $v$ has only one child $u$, then $\mathcal{K}(u) = \mathcal{K}(v)$;

3. If $v$ has two children $u_1, u_2$, we assign the sets to $u_1, u_2$ being an arbitrary partition of $\mathcal{K}(v)$ of *equal* sizes, i.e., $|\mathcal{K}(u_1)| = |\mathcal{K}(u_2)|$, $\mathcal{K}(u_1) \cap \mathcal{K}(u_2) = \emptyset$ and $\mathcal{K}(u_1) \cup \mathcal{K}(u_2) = \mathcal{K}(v)$.

Clearly, the value $\mathcal{K}(v)$ for any node $v$ at level $t$ can be determined with only the realization of $\mathbf{x}^t$ and the values of $\mathcal{K}$ of all nodes at level $t$ form a partition of $K$. The procedure $\mathcal{K}$ fails if there exists some node $v$ with two children such that $|\mathcal{K}(v)| < 2$. Suppose the procedure $\mathcal{K}$ does not fail. We have for any $k \in K$, there exists a unique path $v_1 \to v_2 \to \cdots \to v_{T+1}$ with $v_1$ being the root, such that for all $t \leq T+1$ we have $k \in \mathcal{K}(v_t)$. For any such $k$, we assign the value of $g_k$ on $\mathbf{x}^t$ as:

$$g_k(\mathbf{x}^t) = L(v_t \to v_{t+1}),$$

where $L$ is the edge label function as discussed above. If the procedure $\mathcal{K}$ fails at some node $v_t$, we assign the value of $g_k(\mathbf{x}^j)$ arbitrarily for $j \geq t$.

By definition of the realization tree, for any $h \in \mathcal{H}$ there must be a unique path $v_1 \to \cdots \to v_{T+1}$, with $v_1$ being root such that $h(\mathbf{x}_t) = L(v_t \to v_{t+1})$ for all $t$. Therefore, if the procedure $\mathcal{K}$ does not fail, then for $k \in \mathcal{K}(v_{T+1})$, we have $h(\mathbf{x}_t) = g_k(\mathbf{x}^t)$ for all $t \leq T$ by definition of $g_k$. We now show that by setting $M = \lceil 5\mathbf{Star}(\mathcal{H}) + \log(1/\delta) \rceil$, w.p. $\geq 1 - \delta$ over $\mathbf{x}^T$, the procedure $\mathcal{K}$ will not fail, thus proving that the class $\mathcal{G} = \{g_k : k \in K\}$ is a stochastic sequential covering of $\mathcal{H}$ with confidence $\delta$. To see this, we note that the procedure $\mathcal{K}$ fails at node $v_t$ at level $t$ if and only if there are $\geq M+1$ nodes with two children in the (unique) path $v_1 \to \cdots \to v_t$, where $v_1$ is root, since only rule 3 will reduce the size of value of $\mathcal{K}$ by $1/2$. Assume now the procedure $\mathcal{K}$ fails at node $v_t$. Let $h \in \mathcal{H}$ be a function such that $h(\mathbf{x}_i) = L(v_i \to v_{i+1})$ for all $i \leq t$, which must exist by definition of realization tree. Since any node $v_j$ in the path $v_1 \to \cdots \to v_t$ with two children implies $\mathbf{x}^{j-1}$ *does not* certify $\mathbf{x}_j$ under $h$, we have that there are at least $M+1$ positions $j$ (with $j \leq t$) such that $\mathbf{x}^{j-1}$ does not certify $\mathbf{x}_j$ under $h$. By Lemma 4 and selection of $M$, this happens with probability $\leq \delta$. This completes the proof.

## C   Proof of Theorem 6

The proof will incorporate the SOA argument as in Ben-David et al. (2009) and the result from Theorem 5. For notational convenience, we denote $d = \mathsf{SL}(s) + 1$. For any $I \subset [T]$ with $|I| \leq d$, we will construct a set $\mathcal{G}_I$. Let $\Phi$ be the SOA predictor (similar to Ben-David et al. (2009, Algorithm 1)) that predicts the label for which the remaining consistent subclass has maximum Star-Littlestone dimension at star scale $s$, if both subclasses have $\mathsf{SL}$ dimension 0 we predict the label for which the remaining consistent subclass has maximum Star number (and break ties arbitrarily). We now construct functions in $\mathcal{G}_I$ as follows. The predictions of functions in $\mathcal{G}_I$ are partitioned into 2 phases (start with phase 1). At phase 1, all the functions in $\mathcal{G}_I$ use the same prediction rule as in Lemma 2, that is, if we are at time step $t \in I$, we predict using $1 - \Phi$, else we use $\Phi$ to predict, where $\Phi$ is the SOA prediction rule described above. We enter phase 2 if the remaining consistent class has Star number upper bounded by $s$; we then construct the prediction functions in $\mathcal{G}_I$ as in Theorem 5 with $\mathbf{Star}(\mathcal{H}) = s$, confidence $\delta/T^{d+1}$ and $|\mathcal{G}_I| \leq e^{5s \log T + \log(T^{d+1}/\delta)}$. The covering class $\mathcal{G}$ is defined to be:

$$\mathcal{G} = \bigcup_{I \subset [T],\, |I| \leq d} \mathcal{G}_I.$$

By Theorem 5 with $\mathbf{Star}(\mathcal{H}) = s$ and $\delta = \delta/T^{d+1}$ and computing the number of $I$s, we have

$$|\mathcal{G}| \leq T^{d+1} e^{5s \log T + \log(T^{d+1}/\delta)} \leq e^{O(\max\{d,s\} \log T + \log(1/\delta))}.$$

We now show that $\mathcal{G}$ is indeed a stochastic sequential covering of $\mathcal{H}$ with confidence $\delta$. Let $\mathcal{H}_I$ be the (*random*) subclass of functions in $\mathcal{H}$ that are consistent with $\Phi$ with error pattern $I$ before entering phase 2 [7] (it is possible that $h$ remains on phase 1 until time $T$). Note that all functions in $\mathcal{H}_I$ agree on samples at phase 1. Note also that, with probability 1 we have $\mathcal{H} = \bigcup_{I \subset [T], |I| \leq d} \mathcal{H}_I$. To see this, we note that if $h$ disagreed with the SOA then the remaining consistent class has $\mathsf{SL}(s)$ decreased by at least 1 (similar to the argument as in Ben-David et al. (2009, Lemma 10)) or has Star number $\leq s$ if the current consistent class has $\mathsf{SL}(s) = 0$. This implies that any $h \in \mathcal{H}$ can be disagreed with SOA at most $d$ times before entering phase 2, which must be in some $\mathcal{H}_I$ with $|I| \leq d$. Now, for any $I$ with $|I| \leq d$ we need to show that:

$$\Pr[\mathcal{G}_I \text{ covers } \mathcal{H}_I] \geq 1 - \frac{\delta}{T^{d+1}}.$$

Note that the main difficulty here is that $\mathcal{H}_I$ is a *random* subset. We show that conditioning on any realization of $\mathcal{H}_I$, the above inequality holds (the inequality will then hold by law of total probability). This follows from Theorem 5 by noticing that the samples in phase 2 are still *i.i.d.* and independent of samples in phase 1, and $\mathcal{G}_I$ trivially covers $\mathcal{H}_I$ in phase 1 by definition of $\mathcal{G}_I$ and $\mathcal{H}_I$. The theorem will now follow by a union bound on all the $I$s.

## D  Example to Corollary 6

We give an example here that demonstrates that there exists $\mathcal{H}$ such that $\tilde{r}_T(\mathcal{H}, \mathcal{P}) \leq (1 - e^{-1}) r_T^*(\mathcal{H})$ under Log-loss, where $\mathcal{P}$ is the class of all *i.i.d.* distributions over the domain. Let $\mathcal{H}$ the class of all the functions map $[T] \to \{0, 1\}$. Clearly, we have, by computing Shtarkov sum (Wu et al., 2022b), that:

$$r_T^*(\mathcal{H}) = T.$$

We now provide a specific strategy that achieves smaller $\tilde{r}_T$. The strategy goes as follows, if we haven't seen the sample $\mathbf{x}_t$ then predict $\frac{1}{2}$, else we predict $\frac{y_i + 1/T}{1 + 2/T}$ where $y_i \in \{0, 1\}$ is the true label observed for previous appearance of $\mathbf{x}_t$ (if there are multiple appearance choose arbitrary one). We now observe that the true labels must be consistent with some function in the class. Otherwise the regret will be negative infinite since the functions in $\mathcal{H}$ are $\{0, 1\}$-valued. We now observe that for any distribution over $[T]$ the expected number of distinct elements with $T$ *i.i.d.* samples is upper bounded by

$$(1 - e^{-1})T.$$

Clearly, this is the bound achieved by uniform distribution over $[T]$. To see that this hold for arbitrary distribution as well, we observe the expected number of distinct elements equals:

$$\sum_{t=1}^{T} 1 - (1 - p_t)^T,$$

where $p_t$ is the probability mass of the distribution on sample $t$. We now observe that $(1 - x)^T$ is convex for all $T$ and $x \in [0, 1]$, therefore by Jensen's inequality we have

$$\sum_{t=1}^{T} (1 - p_i)^T \geq T(1 - 1/T)^T \sim Te^{-1}.$$

Since the regret is upper bounded by the expected number of distinct elements, we have

$$\tilde{r}_T(\mathcal{H}, \mathcal{P}) \leq (1 - e^{-1})T + O(1).$$

## E  Stochastic sequential cover for finite Pseudo-dimensional classes

Let $\mathcal{H} \subset [0, 1]^{\mathcal{X}}$ be a real valued function class of Pseudo-dimensional $\mathsf{P}(\mathcal{H})$. We show in this Appendix that there exists a stochastic sequential covering set $\mathcal{G}$ of $\mathcal{H}$ w.r.t. *i.i.d.* processes at scale $\alpha$ and confidence $\delta$ such that

$$\log |\mathcal{G}| \leq O(\mathsf{P}(\mathcal{H}) \log^2(T/\alpha) + \log(T/\alpha) \log(1/\delta)).$$

---

[7] Here, phase 1 and 2 corresponds to that the functions in $\mathcal{H}$ consistent with $h$ on current sample has Star number $> s$ and $\leq s$, respectively.

The proof follows a similar path as in the proof of Theorem 4 but replacing the one-inclusion graph algorithm with the *multi-class* one-inclusion graph algorithm as in Rubinstein et al. (2006). To do so, we denote by $J \subset [0, 1]$ a uniform discretization of $[0, 1]$ with step size $2\alpha$, and $N := |J| \leq \frac{1}{2\alpha}$. Let $\mathcal{H}' = \{h'(\mathbf{x}) = \arg\min_{a \in J}\{|h(\mathbf{x}) - a|\} : h \in \mathcal{H}\}$ be the discretized class of $\mathcal{H}$. Clearly, $\mathsf{P}(\mathcal{H}') \leq \mathsf{P}(\mathcal{H})$. Let $\Phi : (\mathcal{X} \times J)^* \times \mathcal{X} \to J$ be the *multi-class* one-inclusion graph algorithm for $\mathcal{H}'$ as in Rubinstein et al. (2006). We have by Rubinstein et al. (2006, Theorem 5.2)

$$\sup_{\mathbf{x}^t \in \mathcal{X}^t} \sup_{h \in \mathcal{H}'} \mathbb{E}_\sigma \left[ 1\{\Phi(\mathbf{x}^{\sigma(t)}, h(\{\mathbf{x}^{\sigma(t-1)}\})) \neq h(x_{\sigma(t)})\} \right] \leq \frac{\mathsf{P}(\mathcal{H})}{t},$$

where $\sigma$ is uniform random permutation over $[t]$. Invoking Lemma 1 for all $h \in \mathcal{H}'$ and $\mathbf{x}^T \in \mathcal{X}^T$

$$\Pr_{\sigma_T} \left[ \sum_{t=1}^T 1\{\Phi(\mathbf{x}^{\sigma_T(t)}, h(\{\mathbf{x}^{\sigma_T(t-1)}\})) \neq h(\mathbf{x}_{\sigma_T(t)})\} \geq 4\mathsf{P}(\mathcal{H}) \log T + \log(1/\delta) \right] \leq \delta.$$

By generalized Sauer's lemma (Haussler & Long, 1995, Corollary 3), for any $\mathbf{x}^T$ the number of functions of $\mathcal{H}'$ restricted on $\mathbf{x}^T$ is upper bounded by $(TN)^{\mathsf{P}(\mathcal{H})}$. Taking $\delta := \delta/(TN)^{\mathsf{P}(\mathcal{H})}$ and applying union bound, we have

$$\Pr_{\sigma_T} \left[ \sup_{h \in \mathcal{H}'} \sum_{t=1}^T 1\{\Phi(\mathbf{x}^{\sigma_T(t)}, h(\{\mathbf{x}^{\sigma_T(t-1)}\})) \neq h(\mathbf{x}_{\sigma_T(t)})\} \geq 5\mathsf{P}(\mathcal{H}) \log(TN) + \log(1/\delta) \right] \leq \delta.$$

By symmetries of $i.i.d.$ process, for any distribution $\mu$ over $\mathcal{X}$ we find

$$\Pr_{\mathbf{x}^T \sim \mu} \left[ \sup_{h \in \mathcal{H}'} \sum_{t=1}^T 1\{\Phi(\mathbf{x}^t, h(\{\mathbf{x}^{t-1}\})) \neq h(\mathbf{x}_t)\} \geq 5\mathsf{P}(\mathcal{H}) \log(TN) + \log(1/\delta) \right] \leq \delta.$$

We now use a similar argument as in Lemma 2 to construct the sequential covering set. Let $\mathbf{err} = 5\mathsf{P}(\mathcal{H}) \log(TN) + \log(1/\delta)$, $I \subset [T]$ with $|I| \leq \mathbf{err}$ and $K = \{k_t\}_{t \in I} \in J^{|I|}$. For any such $I, K$, we construct a sequential function $g_{I,K}$ such that for any $t \in I$ and $\mathbf{x}^t$ we set $g_{I,K}(\mathbf{x}^t) = k_t$ and set $g_{I,K}(\mathbf{x}^t) = \Phi(\mathbf{x}^t, \{g_{I,K}(\mathbf{x}^{t-1})\})$ for $t \notin I$. It is easy to verify that the class $\mathcal{G}$ consisting of all such $g_{I,K}$ is stochastic sequential cover of $\mathcal{H}'$ at scale 0 and confidence $\delta$, and

$$\log |\mathcal{G}| = \sum_{i=1}^{\mathbf{err}} \binom{T}{i} N^i \leq (TN)^{\mathbf{err}+1} \leq O(\mathsf{P}(\mathcal{H}) \log^2(TN) + \log(TN) \log(1/\delta)).$$

By the construction of $J$, $\mathcal{G}$ is a stochastic sequential covering set of $\mathcal{H}$ w.r.t. $i.i.d.$ processes at scale $\alpha$ and confidence $\delta$. The claimed bound now follows by noticing that $N \leq \frac{1}{2\alpha}$.

# F An $\Omega(\log T)$ lower bound of realizable cumulative error

We now show that there exist classes $\mathcal{H}$ of finite VC-dimension such that for any prediction rule, the expected cumulative error with $i.i.d.$ sampling in the realizable case is lower bounded by $\Omega(\mathsf{VC}(\mathcal{H}) \log T)$. This result was shown in Haussler et al. (1994, Section 3) and also in Antos & Lugosi (1998), however, we provide an alternative (simpler) proof here for completeness.

Let $\mathcal{H} = \{h_a(x) = 1\{x \geq a\} : x, a \in [0, 1]\}$ be the class of all linear threshold functions $[0, 1] \to \{0, 1\}$. We show that $\mathcal{H}$ is the desired class. To do so, we define $\nu$ to be the uniform distribution over $[0, 1]$. Let $A, X_1, \cdots, X_T$ be $i.i.d.$ samples from $\nu$ [8]. We show that for any prediction rule $\Phi$ we have

$$\mathbb{E} \left[ \sum_{t=1}^T 1\{\Phi(X_1^t, h_A(X_1^{t-1})) \neq h_A(X_t)\} \right] \geq \Omega(\log T).$$

W.l.o.g., we can assume that $A, X_1, \cdots, X_T$ are distinct. By symmetries of $i.i.d.$, we may assume that $A, X_1, \cdots, X_T$ is a uniform random permutation for some fixed *set* $\{A, X_1, \cdots, X_T\}$. We consider the following process: at the

---

[8]We only prove the case when $\mathsf{VC}(\mathcal{H}) = 1$, the extension to general VC-dimension $d$ is straightforward by considering a product class $\mathcal{H} = \{h_\mathbf{a}(x, b) = 1\{x \geq a_b\} : \mathbf{a} \in [0, 1]^d, (x, b) \in [0, 1] \times [d]\}$ and defining the distribution to first sample from $[d]$ uniformly and then sample from $\nu$. This admits a lower bound $\Omega(d \log(T/d))$ since w.h.p. each $b \in [d]$ appears $\Theta(T/d)$ times.

beginning Nature randomly permutes $\{A, X_1, \cdots, X_T\}$ and maintains a set $S$ initially to be equal to $\{X_1, \cdots, X_T\}$. At each time step $t$, Nature reveals the relative position of $X_t$ in $S$. After the predictor has made the prediction, Nature discards elements in $S$ on the opposite side of $X_t$ relative to $A$ (i.e., if $A > X_t$ we discard elements $\leq X_t$ in $S$, else we discard elements $\geq X_t$) and reveals them to the predictor. Clearly, a lower bound for the game described above implies a lower bound for the original game, since the predictor gains more information at each time step. Denote $f(t)$ to be the expected number of errors the predictor will make if $|S| = t$. We have the following recursion:

$$f(T) \geq \frac{2}{T} \sum_{t=1}^{T/2} \frac{t \cdot f(t)}{T} + \frac{(T-t) \cdot f(T-t)}{T} + \frac{t}{T} \geq \frac{2}{T} \sum_{t=1}^{T-1} \frac{t \cdot f(t)}{T} + \frac{t}{T}.$$

The reasoning goes as follows: conditioned on the event that $X_1$ is at position $t$, we have probability $t/T$ that $A$ is less than $X_1$ and probability $(T-t)/T$ that $A$ is larger than $X_1$. The best strategy for the predictor is to predict $A > X_1$ if $t \leq T/2$ and predict $A < X_1$ otherwise. This contributes an expected error at first step to be $\frac{t}{T}$. The recursive formula now follows by the observation that conditioning on the position of $X_1$ and relative position of $A$ to $X_1$, the remaining set $S \cup \{A\}$ is still a uniform random permutation. We now claim that (the constant $0.01$ is not optimized):

$$f(t) \geq 0.01 \cdot \log t.$$

The base case for $t = 1$ can be verified easily. We now prove by induction. We observe that

$$\int x \cdot \log(x) = \frac{1}{2} x^2 \cdot \log(x) - \frac{x^2}{4}.$$

Therefore, using Euler–Maclaurin formula we have:

$$T^2 \cdot f(T) \geq 2 \sum_{t=1}^{T-1} 0.01 \cdot t \log(t) + t \geq 0.01 \cdot T^2 \log T - 0.02 \cdot T \log T - 0.01 \cdot T^2/4 + T^2/2 \geq 0.01 \cdot T^2 \log T.$$

The result follows.

