# OpenReview forum: "Expected Worst Case Regret via Stochastic Sequential Covering"
_TMLR — Accepted by TMLR_

### Review · Reviewer_pHFt · 2023-04-10

**Summary Of Contributions:**

The authors address the problem of sequential prediction and online minimax regret with stochastically generated features under a general loss function. They contribute by:

- Introducing the notion of expected worst-case minimax regret, which generalizes and encompasses prior known minimax regrets.
- Establishing upper bounds for such minimax regrets using a novel concept called stochastic global sequential covering.
- Introducing a new complexity measure, the Star-Littlestone dimension, to improve the bound on the cardinality of stochastic global sequential covering.
- Establishing upper bounds for real-valued classes with finite fat-shattering numbers.
- Providing lower bounds for expected worst-case minimax regret by applying information-theoretic tools.

**Audience:**

Yes

**Claims And Evidence:**

Yes

**Requested Changes:**

Is it possible to argue that the newly introduced complexity measures represent the intrinsic difficulty of the problem?  For example, can you show regret lower bound depending on the complexity measures? Or can you provide examples where the regret upper bound provided in the paper is tight (including factors depending on $\mathcal{H}$)?

**Strengths And Weaknesses:**

Strong aspects of the submission:

- Introducing the notion of expected worst-case minimax regret, generalizing and encompassing prior known minimax regrets.
- Proposing the novel concept of stochastic global sequential covering to establish tight upper bounds for minimax regrets.
- Introducing the new complexity measure, the Star-Littlestone dimension, to improve the bound on the cardinality of stochastic global sequential covering.
- The paper is well-structured, and its claims are supported by convincing mathematical arguments.

Weaknesses of the submission:

The submission does not seem to have any significant weaknesses. However, it might be beneficial for the authors to:

- Discuss possible limitations of the method and potential avenues for future research to address those limitations.
- Discuss the gap between the VC dimension and the newly introduced complexity measure (e.g., through concrete examples like Example 6).

---

### Review · Reviewer_tdCv · 2023-06-03

**Summary Of Contributions:**

This paper studies the worst-case regret bound in a contextual bandit problem with a general prediction class. The primary motivation is to understand the scenario where the contexts are revealed via a fixed distribution, even though the reward can still be adversarial. The minimax regret, $r$, defined in this paper is determined by the "regular regret" under the optimal predictor. The paper demonstrates that the regret can be upper-bounded by the stochastic covering number of the hypothesis. The majority of the paper then attempts to upper-bound the covering number using different combinatorial dimensions of the hypothesis class, $H$. The first bound is determined by the VC dimension, such that the log-covering number bounds by VC*$\log(T)^2$. The quadratic dependence on $\log T$ is further improved by introducing a dimension known as the star-Littlestone dimension, $SL$. The paper shows that the log-covering number can be upper-bounded by $SL*\log T$. A lower bound for this quantity is also provided in the pap

**Audience:**

Yes

**Claims And Evidence:**

Yes

**Requested Changes:**

There are some minor editorial issues that should be addressed:

- In the introduction, it would be beneficial to provide more motivation for defining and studying the problem, especially when introducing the concept of minimax regret.
- In the introductory paragraph, the term 'y_t … assume to be convex' is used. Could you please clarify what you mean by 'convex' in this context?
- In the second paragraph of the introduction, you mention that 'H admits a predictor with cumulative error B'. It's not clear what you mean by 'predictor' and 'cumulative error' here.
- It would be helpful to include explanations in layman's terms when introducing new concepts. For instance, when defining the 'star' number, could you also provide a simplified explanation of what it means?
- The definition of the SL dimension should be included in the main body of the paper, rather than being relegated to an appendix or footnote.


**Strengths And Weaknesses:**

The most significant weakness in this paper lies in the motivation for the study. As I understand, the minimax regret $r$ does not measure the regret of any particular algorithm — rather, it represents the hypothetical regret of the best possible algorithm. In this regard, $r$ provides a regret lower-bound for any conceivable algorithm. Therefore, studying the upper bound of this quantity appears illogical as it doesn't convey meaningful information about a specific algorithm. Although it might shed some light on the complexity of the hypothesis class under an online decision-making setting, the insights are not particularly significant. Moreover, the lower bound of the quantity, which arguably carries more weight, is not expressed in terms of the VC, SL dimension, etc. — this omission significantly weakens the paper. For example, if the paper demonstrated that both the lower and upper bounds were given by $c*VC * \log^2(T)$, it would have appeared much stronger.

---

### Review · Reviewer_b28u · 2023-06-09

**Summary Of Contributions:**

This paper considers online learning and sequential prediction. The authors introduce a new minimax notion of regret, the expected worst-case minimax regret, which generalizes other existing notions of minimax regret by taking a supremum over all distributions over the sequence of features observed by the algorithms. The authors then establish upper bounds for this minimax regret using stochastic global sequential covers - which generalize the classical sequential cover. To further bound the regret, they bound the stochastic sequential cover for classes with finite VC dimension, finite star number, and finite Star-Littlestone dimension. This last notion is novel, and the authors demonstrate classes where the Star-Littlestone is meaningful even when the star number and littlestone dimension are infinite. The authors finally provide lower bounds on the minimax regret.


**Audience:**

Yes

**Claims And Evidence:**

Yes

**Requested Changes:**

I think if the authors could more clearly clarify their precise technical contributions it would be easier for a reader to understand the contribution.

**Strengths And Weaknesses:**


Overall the results in the paper seem comprehensive, and the authors have tried to interpret their notion of stochastic sequential covering for many of the most common classes contained in the literature including finite VC-dimension, finite Star-dimension, and finite fat-shattering.

Here are some concrete comments/concerns.

Definition of Expected Worst Case Minimax Regret Equation (1): I’m sure this is extremely naive, but I am confused about what this definition adds. In general, the sup of the expectation over all distributions should occur at a single sequence. Thus I would expect that if $P$ is all distributions over $X^T$, that equations (1) and (3) would be the same.  In fact, (1) and (3) should be the same for any class P which contains all singleton distributions. As the authors remark, (3) is well studied in the literature. Can the authors remark on examples of $P$ that may be interesting that don’t contain the singleton distributions? Are i.i.d processes the natural example?

Stylistic Comment: The paper is extremely thorough, but as a result, somewhat challenging to read through. In addition, from a reviewing perspective, it’s hard to tell what the contributions of the authors. Here are some specific comments on this:
- What is novel in Theorem 1? Presumably existing works have a very similar analysis?
- Can the authors comment on whether existing results for classical minimax regret (e.g Cesa-Bianchi Lugosi ’06) recover the results of Corollary 1?
- Maybe some of the technical lemmas and their proofs can be in the appendix? Eg Lemma 2,3,4 and the proof of Theorem 4. I don’t know what the 1-inclusion graph predictor is and so reading the proof was not helpful at all.
- Along these lines, I felt there were places where things were not defined sufficiently clearly. For example, in Lemma 4 - what is Phi?

---

### Decision · Action_Editors · 2023-08-01

**Recommendation:** Accept as is

**Comment:**

The paper studies the online minimax regret of sequential prediction with stochastically generated features.  By considering a relaxed adversarial setting and introducing novel theoretical concepts, the authors derive significantly tighter bounds than the known bounds for adversarial settings.

Reviewers originally mentioned the following pros and cons:

Pros:
- The theoretical results are comprehensive with novel concepts introduced.
- Derived significantly tighter bounds than known results (under a different setting).

Cons:
- Weak motivation (deriving upper-bounds of lower-bounds might not be useful)
- Unclear contributions

The authors addressed concerns appropriately and revised the manuscript, which improved the readability.
All reviewers say yes for the two TMLR criteria, and most of them recommend acceptance.

For the camera-ready version, please use the "citet" command (remove parentheses) for references which are parts of sentences.


**Audience:**

yes

**Claims And Evidence:**

yes